EMBO
Molecular Medicine

# IgE actions on CD4+ T cells, mast cells, and macrophages participate in the pathogenesis of experimental abdominal aortic aneurysms

Jing Wang[1], Jes S Lindholt[2], Galina K Sukhova[1], Michael A Shi[1], Mingcan Xia[3], Han Chen[1,4], Meixiang Xiang[4], Aina He[1], Yi Wang[1], Na Xiong[3], Peter Libby[1], Jian-An Wang[4],** & Guo-Ping Shi[1],*

## Abstract

Immunoglobulin E (IgE) activates mast cells (MCs). It remains unknown whether IgE also activates other inflammatory cells, and contributes to the pathogenesis of abdominal aortic aneurysms (AAAs). This study demonstrates that CD4+ T cells express IgE receptor FcεR1, at much higher levels than do CD8+ T cells. IgE induces CD4+ T-cell production of IL6 and IFN-γ, but reduces their production of IL10. FcεR1 deficiency (Fcer1a−/−) protects apolipoprotein E-deficient (Apoe−/−) mice from angiotensin-II infusion-induced AAAs and reduces plasma IL6 levels. Adoptive transfer of CD4+ T cells (but not CD8+ T cells), MCs, and macrophages from Apoe−/− mice, but not those from Apoe−/− Fcer1a−/− mice, increases AAA size and plasma IL6 in Apoe−/− Fcer1a−/− recipient mice. Biweekly intravenous administration of an anti-IgE monoclonal antibody ablated plasma IgE and reduced AAAs in Apoe−/− mice. Patients with AAAs had significantly higher plasma IgE levels than those without AAAs. This study establishes an important role of IgE in AAA pathogenesis by activating CD4+ T cells, MCs, and macrophages and supports consideration of neutralizing plasma IgE in the therapeutics of human AAAs.

**Keywords** abdominal aortic aneurysm; IgE; macrophage; mast cell; T cells
**Subject Categories** Haematology; Immunology

## Introduction

Mast cells (MCs) contribute importantly to the pathogenesis of atherosclerosis and abdominal aortic aneurysms (AAAs) (Kovanen, 2007; Sun et al, 2007a,b; Bot & Biessen, 2011; Swedenborg et al, 2011). After activation, MCs release pro-inflammatory cytokines (e.g. IFN-γ and IL6) to induce matrix-degrading protease expression

from vascular smooth muscle cells (SMCs) and endothelial cells (ECs) (Sun et al, 2007a,b); MC-specific proteases (e.g. chymase and tryptase) to induce vascular cell apoptosis (Heikkilä et al, 2008; Sun et al, 2009; Zhang et al, 2011; den Dekker et al, 2012); angiogenic factors (e.g. basic fibroblast growth factor) to stimulate angiogenesis (Lappalainen et al, 2004); and histamine to induce EC inflammatory responses (Dileepan & Stechschulte, 2006). Multiple mechanisms—including lipoproteins, immunoglobulins, neurotrophic factors (e.g. substance P and neuron growth factor), stem cell factor, complements (e.g. C3a and C5a), endotoxin, inflammatory cytokines (e.g. IL1 and TNF-α) (Xu & Shi, 2012), or even acute restraint stress (Huang et al, 2002)—can mediate MC activation. Among these MC activators, immunoglobulin E (IgE) is probably the best known (Ishizaka et al, 1978; Razin et al, 1983; Dvorak et al, 1985). Mast cells inactivation by blocking IgE activity by use of anti-IgE antibodies has been investigated widely among patients with allergic asthma, food allergy, atopic dermatitis, atopic eczema, chronic autoimmune urticaria, and chronic rhinosinusitis (Milgrom et al, 1999; Leung et al, 2003; Busse et al, 2011). We recently found that in addition to MCs, IgE also targets macrophages, SMCs, and ECs, and induces cytokine and chemokine production from these atherosclerosis-pertinent cells (Wang et al, 2011). This study demonstrates that T cells express IgE high affinity receptor FcεR1. Using angiotensin-II (Ang-II) infusion-induced experimental AAAs in apolipoprotein E-deficient (Apoe−/−) mice, we tested whether IgE actions on T cells, MCs, and macrophages contribute to AAA pathogenesis and whether inhibition of IgE activity using FcεR1-deficient mice or ablation of plasma IgE using anti-IgE monoclonal antibody (mAb) ameliorates AAA development.

## Results

### FcεR1 expression on CD4+ and CD8+ T cells

Inflammatory cells express the high-affinity IgE receptor FcεR1, e.g. MCs, basophils, eosinophils, dendritic cells, Langerhans cells,

1 Department of Medicine, Brigham and Women's Hospital and Harvard Medical School, Boston, MA, USA
2 Department of Cardiovascular and Thoracic Surgery, Elitary Research Centre of Individualized Medicine in Arterial Diseases, University Hospital of Odense, Odense, Denmark
3 Department of Veterinary and Biomedical Sciences, The Pennsylvania State University, University Park, PA, USA
4 Cardiovascular Key Lab of Zhejiang Province, Department of Cardiology, College of Medicine, The Second Affiliated Hospital, Zhejiang University, Hangzhou, China
*Corresponding author. Tel: +1 617 525 4358; Fax: +1 617 525 4380; E-mail: gshi@rics.bwh.harvard.edu
**Corresponding author. Tel: +86 138 057 86328; Fax: +86 571 87022776; E-mail: wja@zju.edu.cn

macrophages, and monocytes (Baniyash *et al*, 1986; Shibaki *et al*, 1996; Boesiger *et al*, 1998; Dombrowicz *et al*, 2000; Katoh *et al*, 2000; Grayson *et al*, 2007; Mancardi *et al*, 2008). Neurons and vascular SMCs and ECs also express FcεR1 (Andoh & Kuraishi, 2004; Wang *et al*, 2011), suggesting a broad spectrum of target cells for IgE. IgE activates MCs by targeting FcεR1 (Xu & Shi, 2012). We have recently shown that IgE also activates macrophages via interaction with cell surface FcεR1 (Wang *et al*, 2011). This study used CD4[+] and CD8[+] T cells selected from splenocytes from wild-type C57BL/6 mice by nylon wool column purification and consequently anti-I-A[b] antibody depletion (Kokkinopoulos *et al*, 1992), followed by anti-CD4 and anti-CD8 antibody-coated magnetic column positive selection. Cell purity ranged from 94 to 98% (Supplementary Fig S1). To examine whether T cells also express FcεR1, we treated both CD4[+] and CD8[+] T cells with IgE, inflammatory cytokines (IFN-γ, IL6, and TNF-α), or Ang-II. Real-time polymerase chain reaction (RT-PCR) demonstrated the expression of mRNAs that encode all three FcεR1 subunits—FcεR1α, FcεR1β, and FcεR1γ—in CD4[+] T cells, but at much lower levels in CD8[+] T cells. Cell stimulation with IgE, cytokines, or Ang-II did not significantly change FcεR1 expression from either CD4[+] or CD8[+] T cells (Fig 1A). To determine what percentages of these T cells express FcεR1, we gated total CD3[+] T cells from mouse splenocytes and selected CD4[+] and CD8[+] T cells to detect the percentage of FcεR1[+] cells in these two populations of T cells (Supplementary Fig S2). Flow cytometry (FACS) analysis of splenocytes using anti-FcεR1α-biotin mAb demonstrated comparable percentages of CD4[+] and CD8[+] T cells that express surface FcεR1α, but negligible intracellular FcεR1α after subtracting the baseline signals from the antibody isotype controls (Fig 1B). Supplementary Fig S3 shows a representative FACS analysis. To compare the relative expression of the same three FcεR1 subunits in CD4[+] T cells to those of cells known to express FcεR1, we performed RT-PCR from both bone marrow-derived MC (BMMCs) and macrophages and found that BMMCs and macrophages expressed much higher levels of the FcεR1α subunit, but comparable levels of FcεR1β (BMMCs: 0.08–0.17%; macrophages: <0.001–0.025% relative to β-actin) and FcεR1γ (BMMCs: 0.40–0.80%; macrophages: 0.09–1.27% relative to β-actin) subunits to those of CD4[+] T cells, depending on the stimulus (Fig 1C). Immunoblot analysis showed that FcεR1 protein expression was also much higher in CD4[+] T cells than in CD8[+] T cells and that IgE, cytokines, and Ang-II did not affect FcεR1α expression (Fig 1D), concordant the results of RT-PCR and FACS analysis (Fig 1A and B). These data suggest that CD4[+] T cells expressed much higher levels of FcεR1 (RT-PCR) than did CD8[+] T cells, although similar percentages of these T cells expressed FcεR1 (FACS). We have previously shown that either aggregated (SPE-7) or monomeric (H1 DNP-ε-206) forms of IgE have equal potency as macrophage activators, and that IgE antigen did not affect this activity of IgE (Wang *et al*, 2011). To test whether FcεR1 on CD4[+] and CD8[+] T cells was functional and also independent of antibody cross-linking, we demonstrated that SPE-7 IgE produced a concentration-dependent increase in the expression of IL6 and IFN-γ mRNA and protein production by CD4[+] and CD8[+] T cells, as determined by RT-PCR (Fig 1E) and by culture medium ELISA (Fig 1F). In contrast, CD4[+] T cells showed biphasic but insignificant changes in IL10 mRNA levels after exposure to low concentrations of IgE (0–10 µg/ml), but demonstrated significant reduction of IL10 expression under high IgE concentrations (50–100 µg/ml)

(Fig 1E). Cell culture medium ELISA showed that IgE activation dose-dependently reduced CD4[+] T-cell IL10 production (Fig 1F). Consistent with the differences in FcεR1 expression levels between CD4[+] and CD8[+] T cells, cellular IFN-γ mRNA levels and medium IL6 protein levels were significantly higher in CD4[+] T cells than in CD8[+] T cells at various IgE concentrations. In contrast, we detected minimal levels of mRNA and medium IL10 in CD8[+] T cells (Fig 1E and F).

Why IgE promotes MC and basophil survival and proliferation (Kawakami & Galli, 2002), but induces macrophage apoptosis (Wang *et al*, 2011), remains unknown. Altered cytokine expression from IgE-treated CD4[+] and CD8[+] T cells may result from IgE actions affecting T-cell survival or proliferation. To test this possibility, we cultured both CD4[+] and CD8[+] T cells in anti-CD3 (1 µg/ml) and anti-CD28 (1 µg/ml) monoclonal antibodies and stimulated these cells with IgE (50 µg/ml) for 3 days. 3-(4,5-dimethylthiazol-2-yl)-2,5-diphenyltetrazolium bromide (MTT) cell survival assay and medium IL2 ELISA demonstrated that IgE did not affect either survival or proliferation of CD4[+] or CD8[+] T cells from either *Apoe*[−/−] mice or FcεR1a-deficient *Apoe*[−/−] *Fcer1a*[−/−] mice (Supplementary Fig S4A–D).

## Absence of FcεR1 reduces experimental AAAs in *Apoe*[−/−] mice

Chronic subcutaneous infusion of Ang-II induces experimental AAAs in *Apoe*[−/−] mice (Daugherty *et al*, 2000; Schulte *et al*, 2010; Zhang *et al*, 2012). We first tested whether AAA development in *Apoe*[−/−] mice associates with plasma IgE level changes. Among 16 *Apoe*[−/−] mice that received Ang-II infusion, 13 developed AAAs and 3 did not. Mice with confirmed AAAs had significantly higher plasma IgE levels than those before Ang-II infusion or those that did not develop AAA even after receiving Ang-II (Fig 2A), although their blood pressures or heart rates did not differ significantly (Supplementary Fig S5). Increased plasma IgE may trigger inflammatory cytokine expression from T cells, MCs, macrophages, and other inflammatory cells, thereby promoting AAA pathogenesis. Interruption of IgE activity may thus attenuate AAA growth. To test this hypothesis, we produced Ang-II infusion-induced AAAs in both FcεR1-sufficient *Apoe*[−/−] mice and FcεR1-deficient *Apoe*[−/−] *Fcer1a*[−/−] mice. In these experiments, *Apoe*[−/−] mice tended to hvae a higher post-Ang-II infusion mortality rate than that of *Apoe*[−/−] *Fcer1a*[−/−] mice (40% versus 20%, *P* = 0.314, Fig 2B). To achieve sufficient numbers of mice that survived from Ang-II infusion and for AAA lesion analysis, we used 30 *Apoe*[−/−] mice and 15 *Apoe*[−/−] *Fcer1a*[−/−] mice to provide 18 *Apoe*[−/−] and 12 *Apoe*[−/−] *Fcer1a*[−/−] survivors after 28 days of Ang-II infusion. *Apoe*[−/−] *Fcer1a*[−/−] mice showed a much lower AAA incidence than did *Apoe*[−/−] mice (33.3% versus 83.3%, *P* = 0.008, Fig 2B). Suprarenal maximal outer aortic diameters measured *in situ* from anesthetized mice were significantly smaller in *Apoe*[−/−] *Fcer1a*[−/−] mice than in *Apoe*[−/−] mice (Fig 2B). As we anticipated, *Apoe*[−/−] and *Apoe*[−/−] *Fcer1a*[−/−] mice that developed AAAs had significantly higher plasma IgE levels than did those did not develop AAAs, although FcεR1-deficiency enhanced plasma IgE levels post-Ang-II infusion, independent of AAA formation. In contrast, plasma IgG or IgM levels did not change significantly in mice with and without AAAs from either genotype (Fig 2C). AAA lesion inflammation (macrophage-positive area, T-cell content, dendritic cell content, major histocompatibility complex [MHC] class-II content, and chemokine monocyte chemoattractant protein-1 [MCP-1] content), lesion cell proliferation (Ki67-positive areas),

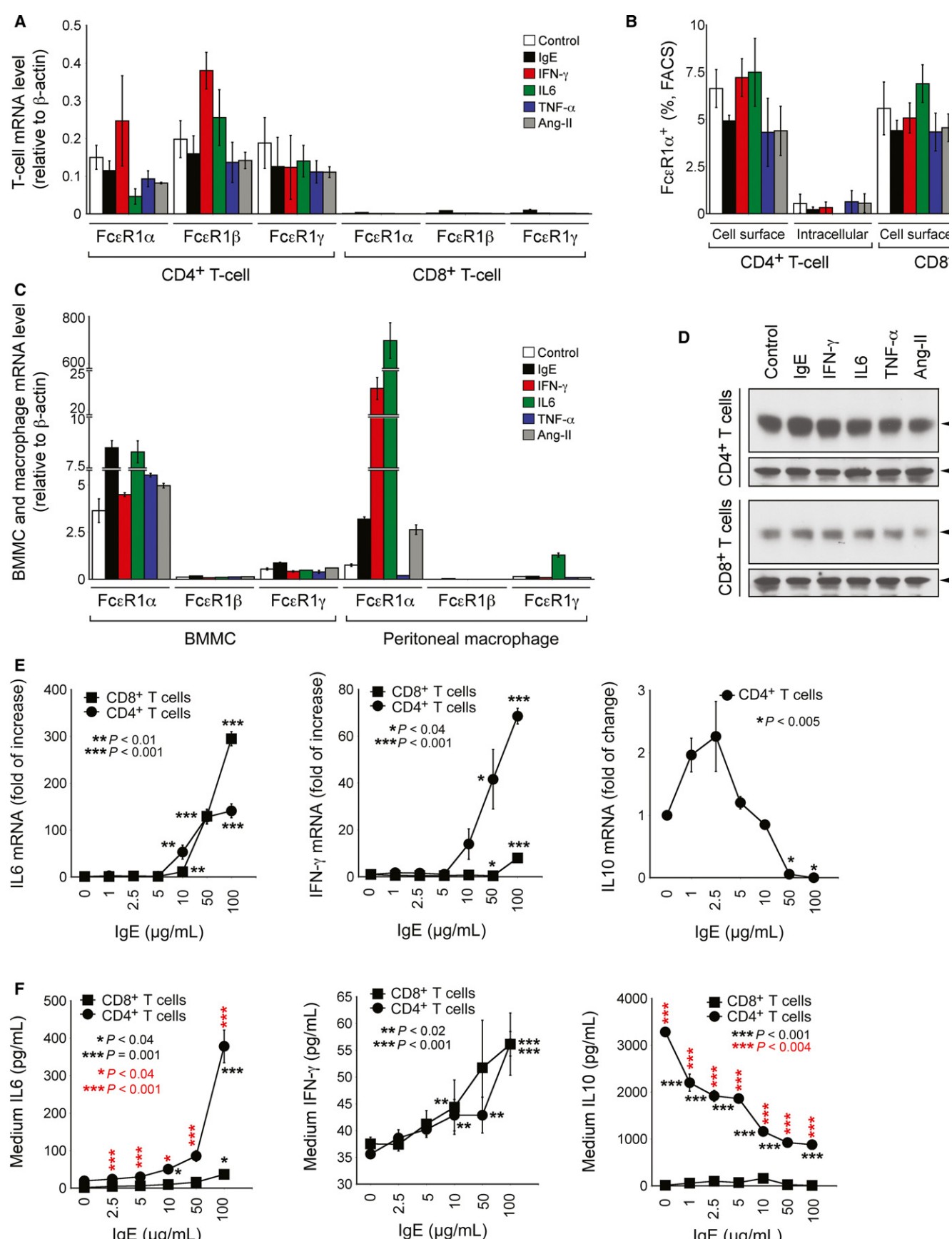

**Figure 1.**

microvessel number (CD31), apoptotic cell number, media elastin fragmentation, and presence of SMC in the tunica media (loss of media SMCs) were all significantly lower in AAA lesions from $Apoe^{-/-}Fcer1a^{-/-}$ mice than in lesions from $Apoe^{-/-}$ mice (Fig 2D and E). However, FcεR1 deficiency did not significantly affect body weight, diastolic and systolic blood pressures, heart rate, plasma lipid profiles (Supplementary Fig S6A–E) before or after AAA production, or AAA lesion total interstitial collagen contents (Supplementary Fig S6F). These observations establish an important role of IgE in AAA pathogenesis.

## IgE actions on CD4+ T cells are essential to AAAs

Increased plasma IgE in AAA mice (Fig 2A and C), reduced AAAs in FcεR1-deficient $Apoe^{-/-}Fcer1a^{-/-}$ mice (Fig 2B), and increased IL6 and IFN-γ and reduced IL10 in CD4+ T cells after IgE activation (Fig 1E and F) suggest a role of IgE actions on CD4+ T cells in AAAs. To test this hypothesis, we adoptively transferred equal numbers ($1 \times 10^7$ per recipient) of splenic CD4+ T cells from age-matched $Apoe^{-/-}$ mice and $Apoe^{-/-}Fcer1a^{-/-}$ mice to $Apoe^{-/-}Fcer1a^{-/-}$ recipient mice, followed by Ang-II infusion to elicit AAA formation. After 28 days, we confirmed the presence of FcεR1a-positive donor cells in AAA lesions of recipient $Apoe^{-/-}Fcer1a^{-/-}$ mice receiving CD4+ T cells from $Apoe^{-/-}$ mice (Fig 3A, left two panels) using a hamster anti-mouse FcεR1a antibody. Donor cells from $Apoe^{-/-}Fcer1a^{-/-}$ mice were used as negative controls for immunostaining (Fig 3A, right panel). Antibody specificity was further tested using AAA lesion consecutive sections with or without the primary antibody and using AAA sections from the $Apoe^{-/-}Fcer1a^{-/-}$ mice (Supplementary Fig S7). Low magnifications (20× or 40×) showing the entire aortas of $Apoe^{-/-}Fcer1a^{-/-}$ recipient mice receiving donor cells from $Apoe^{-/-}$ mice and $Apoe^{-/-}Fcer1a^{-/-}$ mice are illustrated in Supplementary Fig S8A. Immunofluorescent co-staining of AAA lesions from $Apoe^{-/-}Fcer1a^{-/-}$ recipient mice receiving donor CD4+ T cells from $Apoe^{-/-}$ mice for FcεR1a and Ki67 demonstrated that donor CD4+ T cells not only targeted to the AAA lesions, but also underwent clonal expansion (Fig 3B). As in $Cd3e^{-/-}$ mice (Milner et al, 2007), CD4+ T-cell reconstitution increased plasma IgE levels in recipient mice, but we did not detect any differences between the two donor cells from $Apoe^{-/-}$ mice and $Apoe^{-/-}Fcer1a^{-/-}$ mice (Fig 3C). Therefore, CD4+ T-cell-induced plasma IgE level increase did not depend on FcεR1 expression, although the mechanism by which CD4+ T cells augment recipient mice IgE levels remains incompletely understood (Milner et al, 2007). Although not statistically significant, CD4+ T cells from $Apoe^{-/-}$ mice partially or fully restored both the AAA incidence rate and post-Ang-II mortality rate in $Apoe^{-/-}Fcer1a^{-/-}$ reci-

pient mice much greater than those from $Apoe^{-/-}Fcer1a^{-/-}$ mice, even though they had similar levels of plasma IgE (Fig 3C). Suprarenal aortic diameters were also significantly reversed in $Apoe^{-/-}Fcer1a^{-/-}$ mice after receiving CD4+ T cells from $Apoe^{-/-}$ mice, but not those from $Apoe^{-/-}Fcer1a^{-/-}$ mice (Fig 3C). Reconstitution of CD4+ T cells from $Apoe^{-/-}Fcer1a^{-/-}$ mice did not affect recipient mice lesion macrophage contents (Fig 3D, left panel), but lesion total CD4+ T-cell numbers did not differ significantly from those receiving CD4+ T cells from $Apoe^{-/-}$ mice (Fig 3D, middle panel)—suggesting that absence of FcεR1 on donor CD4+ T cells did not affect AAA lesion T-cell infiltration, but reduced macrophage accumulation. Therefore, the FcεR1 on CD4+ T cells may be permissive for macrophage infiltration into AAA lesions, similar to the prior conclusion that M1 macrophage infiltration in white adipose tissues requires prior CD8+ T-cell activation (Nishimura et al, 2009). Reduced macrophages in AAA lesions from recipient mice receiving CD4+ T cells from $Apoe^{-/-}Fcer1a^{-/-}$ mice yielded lower levels of MHC class-II—marker of inflammation (Zhang et al, 2012)—in these AAA lesions, compared with those receiving CD4+ T cells from $Apoe^{-/-}$ mice, although such difference did not reach statistical significance (Fig 3D, right panel).

In contrast, recipient $Apoe^{-/-}Fcer1a^{-/-}$ mice receiving CD8+ T cells from $Apoe^{-/-}$ mice or $Apoe^{-/-}Fcer1a^{-/-}$ mice did not change AAA formation in $Apoe^{-/-}Fcer1a^{-/-}$ recipient mice. Anti-FcεR1a mAb immunostaining and immunofluorescent co-staining together with an anti-Ki67 rabbit polyclonal antibody of AAA lesion sections from $Apoe^{-/-}Fcer1a^{-/-}$ recipient mice receiving donor CD8+ T cells from $Apoe^{-/-}$ mice showed the presence of donor CD8+ T cells in AAA lesions (Supplementary Fig S8B and S9A, left two panels) and these donor cells also underwent clonal expansion (Supplementary Fig S9B). Sections from recipient mice receiving CD8+ T cells from $Apoe^{-/-}Fcer1a^{-/-}$ mice served as negative controls for the FcεR1a mAb (Supplementary Fig S8B and S9A, right panel). Unlike donor CD4+ T cells, donor CD8+ T cells from either $Apoe^{-/-}$ mice or $Apoe^{-/-}Fcer1a^{-/-}$ mice did not increase plasma IgE levels significantly (Supplementary Fig S9C). Although we detected mild increases in AAA incidence in all mice that received donor CD8+ T cells from $Apoe^{-/-}$ mice or $Apoe^{-/-}Fcer1a^{-/-}$ mice, and there were complete reverses of the post-Ang-II mortality rate in mice that received CD8+ T cells from $Apoe^{-/-}$ mice but not from $Apoe^{-/-}Fcer1a^{-/-}$ mice, suprarenal aortic diameters from $Apoe^{-/-}Fcer1a^{-/-}$ mice that received donor cells from $Apoe^{-/-}$ mice or $Apoe^{-/-}Fcer1a^{-/-}$ mice were not different from those of the parental $Apoe^{-/-}Fcer1a^{-/-}$ mice (Supplementary Fig S9C). Consistent with unchanged aortic diameters, AAA lesion macrophage content did not differ between $Apoe^{-/-}Fcer1a^{-/-}$ mice and

---

**Figure 1. FcεR1 expression and activity in cells treated with or without different stimuli.**

A  RT-PCR to determine mRNA levels of three FcεR1 subunits, FcεR1α, FcεR1β, and FcεR1γ relative to β-actin from CD4+ and CD8+ T cells.
B  FACS analysis to determine the percentage of FcεR1α-expressing T cells by staining surface and intracellular FcεR1α on CD4+ and CD8+ T cells after subtracting the background signals from the IgG isotypes.
C  RT-PCR to determine mRNA levels of three FcεR1 subunits relative to β-actin from bone marrow-derived mast cells (BMMCs) and macrophages.
D  Immunoblot analysis of FcεR1α on CD4+ and CD8+ T cells. Both CD4+ and CD8+ T-cell immunoblots were from the same SDS–PAGE.
E  RT-PCR to determine cytokine (IL6, IFN-γ, and IL10) mRNA level changes in CD4+ and CD8+ T cells, treated with or without different doses of IgE (SPE-7) as indicated. *P* values refer to comparisons to untreated cells (IgE = 0 μg/ml).
F  ELISA to determine culture medium cytokine (IL6, IFN-γ, and IL10) levels in CD4+ and CD8+ T cells, treated with and without different doses of IgE as indicated. *P* values in black refer to comparisons to untreated cells (IgE = 0 μg/ml). *P* values in red refer to comparisons between CD4+ and CD8+ T cells.

Data information: Data in panels A-E are mean ± SEM from 3–5 independent experiments.

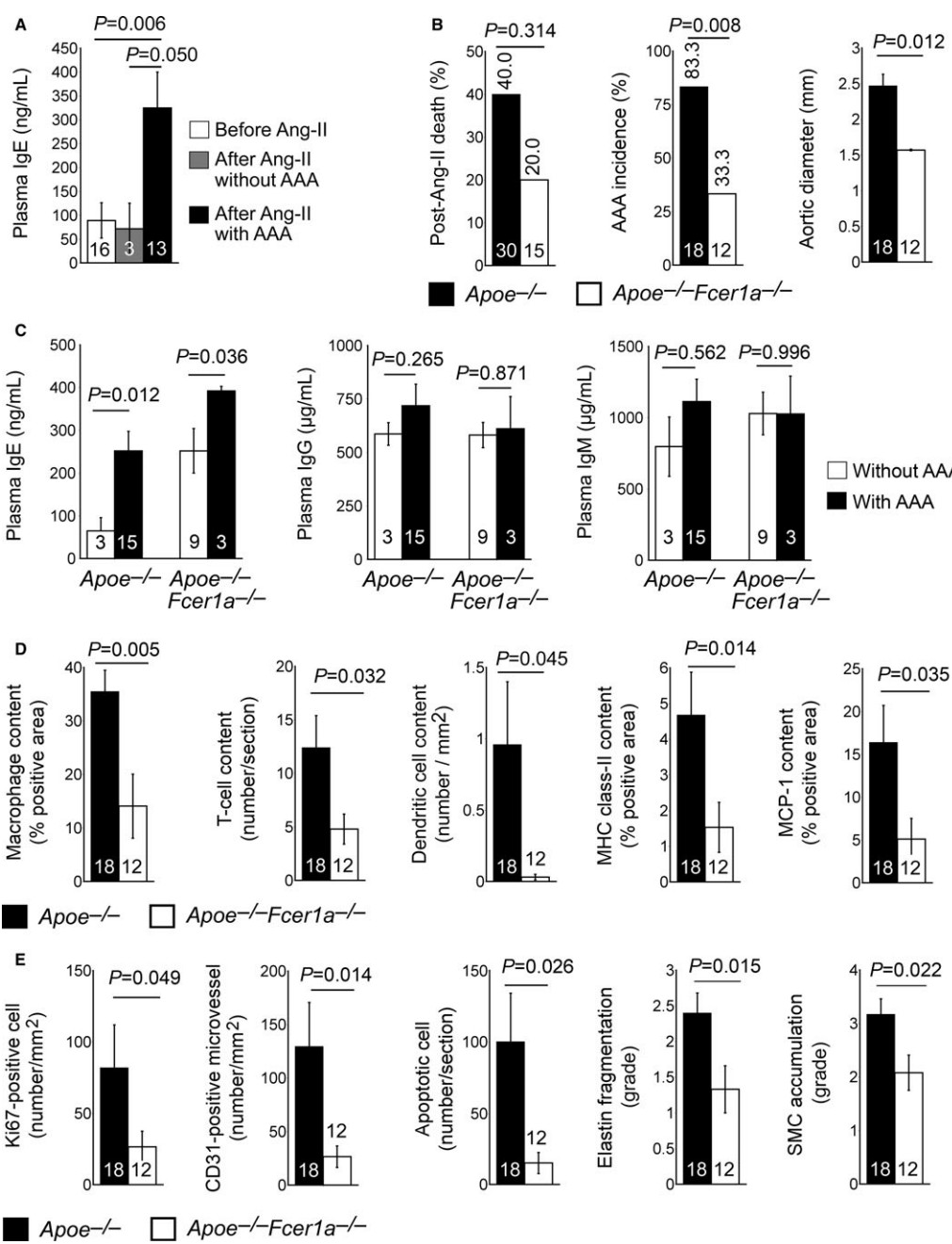

**Figure 2.  Role of FcεR1 in Ang-II infusion-induced AAAs.**

A    ELISA determined plasma IgE levels in *Apoe*[−/−] mice before and 28 days after Ang-II infusion with and without AAA formation.
B    Post-Ang-II infusion rupture-associated mortality rate calculated from survived mice versus total mice used (indicated), and AAA incidence and maximal suprarenal outer aortic diameter from *in situ* calculated from all survived mice.
C    Plasma IgE, IgG, and IgM levels in survived mice with and without AAAs.
D, E   AAA lesion macrophage content, CD4[+] T-cell content, dendritic cell content, major histocompatibility complex (MHC) class-II-positive area, and chemokine MCP-1-positive area (D), and AAA lesion Ki67-positive proliferating cell number, CD31-positive microvessel number, TUNEL-positive apoptotic cell area, arterial wall elastin fragmentation grade, and media smooth muscle cell (SMC) accumulation grade (E) from both *Apoe*[−/−] and *Fcer1a*[−/−]*Apoe*[−/−] mice harvested at 28 days after Ang-II infusion.

Data information: Data are mean ± SEM. Number of mice per group is indicated in each bar.

those receiving CD8[+] T-cell adoptive transfer (Supplementary Fig S9D, left panel), although lesion T-cell content and MHC class-II-positive areas increased in AAA lesions from recipient mice receiving donor cells from *Apoe*[−/−] mice, but did not reach statistical significance from those receiving donor cells from *Apoe*[−/−] *Fcer1a*[−/−] mice (Supplementary Fig S9D, right two panels). Our

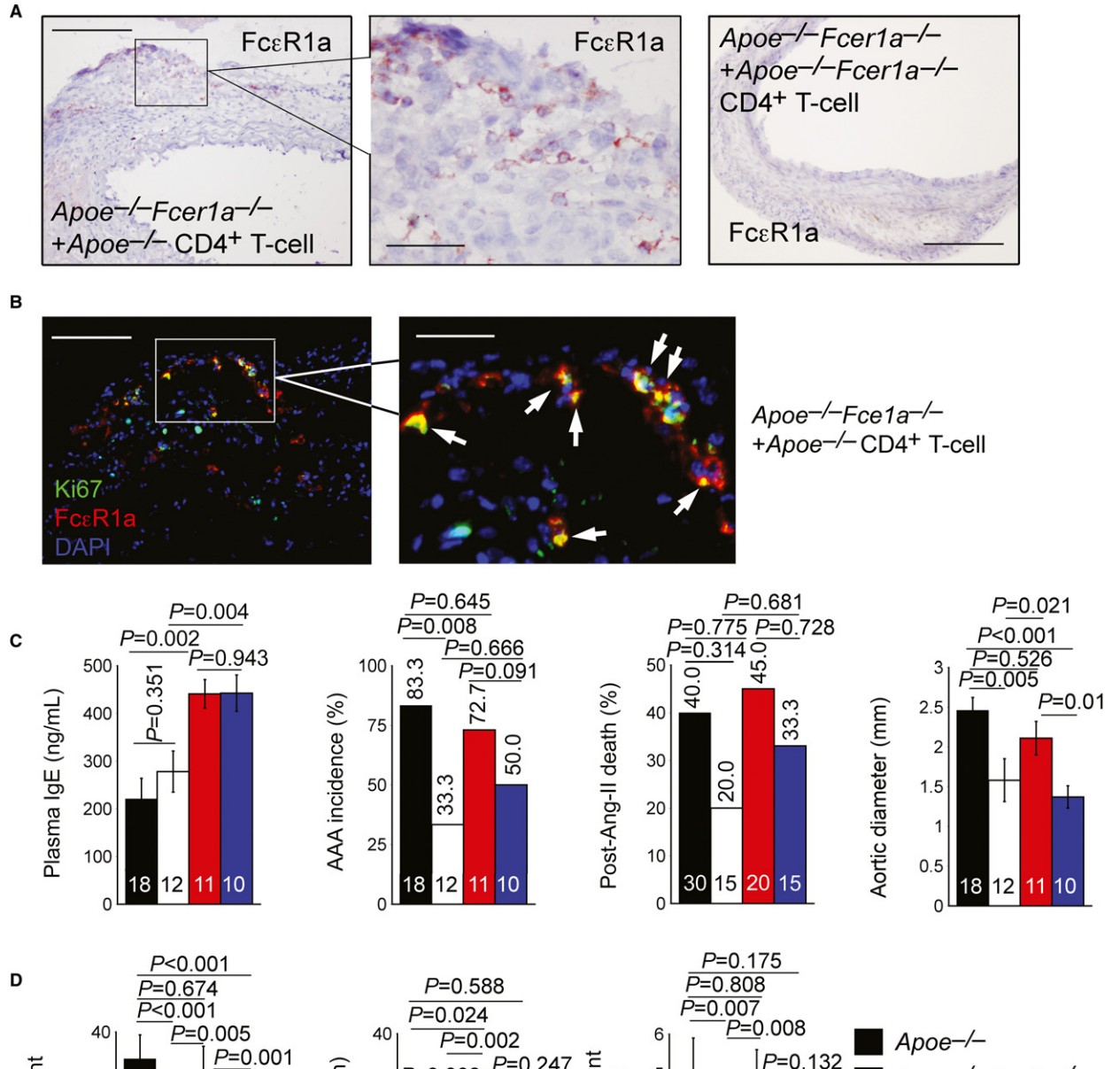

**Figure 3.  IgE actions on CD4+ T cells in AAAs.**

A   Anti-FcεR1a antibody-mediated immunostaining to detect donor CD4+ T cells from *Apoe*−/− mice (left two panels) and *Fcer1a*−/−*Apoe*−/− mice (right panel) in recipient *Fcer1a*−/−*Apoe*−/− mice. Scale: 200 μm; insert scale: 50 μm.

B   FcεR1a and Ki67 antibody-mediated immunofluorescent co-staining to detect proliferating donor CD4+ T cells from *Apoe*−/− mice in recipient *Fcer1a*−/−*Apoe*−/− mice. Scale: 200 μm; insert scale: 60 μm.

C, D   (C) Plasma IgE level, AAA incidence, post-Ang-II infusion aortic rupture-associated mortality rate, and suprarenal maximal outer aortic diameter; and (D) AAA lesion macrophage content, CD4+ T-cell content, and major histocompatibility complex (MHC) class-II-positive area from *Apoe*−/− mice, *Fcer1a*−/−*Apoe*−/− mice, and *Fcer1a*−/−*Apoe*−/− recipient mice receiving donor CD4+ T cells from *Apoe*−/− and *Fcer1a*−/−*Apoe*−/− mice. Data are mean ± SEM. Number of mice per group is indicated in each bar.

observations from $Apoe^{-/-}Fcer1a^{-/-}$ mice reconstituted with CD4[+] and CD8[+] T cells suggest that IgE effects on CD4[+] T cells influenced AAA pathogenesis to a much greater extent than did CD8[+] T cells.

### AAA formation requires IgE actions on MCs and macrophages

The foregoing experiments established an important role of IgE on CD4[+] T cells, but both MCs and macrophages also express FcεR1 (Fig 1C) (Boesiger et al, 1998; Wang et al, 2011). IgE activates MCs and induces MC degranulation and release of granule-associated inflammatory mediators (Ishizaka et al, 1978; Razin et al, 1983; Dvorak et al, 1985). IgE also activates macrophages to release cytokines and chemokines or to undergo apoptosis (Wang et al, 2011). IgE actions on these inflammatory cells therefore may also contribute to AAA pathogenesis as do those from CD4[+] T cells.

To examine the role of IgE activity on MCs in AAAs, we prepared BMMCs from both $Apoe^{-/-}$ and $Apoe^{-/-}Fcer1a^{-/-}$ mice and adoptively transferred them intravenously into $Apoe^{-/-}Fcer1a^{-/-}$ recipient mice, followed by Ang-II infusion to elicit AAA formation. Anti-FcεR1a mAb immunostaining demonstrated the anticipated accumulation of donor FcεR1a-positive BMMCs from $Apoe^{-/-}$ mice in $Apoe^{-/-}Fcer1a^{-/-}$ recipient mouse AAA lesions (Supplementary Fig S8C and Fig 4A, left two panels). The same immunostaining of AAA sections from recipient mice that received BMMCs from $Apoe^{-/-}Fcer1a^{-/-}$ mice tested antibody specificity (Supplementary Fig S8C and Fig 4A, right panel). Similar to results obtained with CD4[+] T-cell-reconstituted $Apoe^{-/-}Fcer1a^{-/-}$ recipient mice, BMMC reconstitution increased recipient mice plasma IgE levels, but did not differ between the two donor cells (Fig 4B). BMMCs from $Apoe^{-/-}$ mice, but not those from $Apoe^{-/-}Fcer1a^{-/-}$ mice, increased AAA incidence, post-Ang-II mortality rate, and maximal outer aortic diameter in $Apoe^{-/-}Fcer1a^{-/-}$ recipient mice (Fig 4B). Statistically insignificant differences in AAA incidences and post-Ang-II mortality might be due to our relatively small sample sizes. AAA lesion apoptotic cell number (Fig 4C) and microvessel number (Fig 4D) were also restored partially in recipient mice receiving BMMCs from $Apoe^{-/-}$ mice, but not receiving those from $Apoe^{-/-}Fcer1a^{-/-}$ mice. Surprisingly, reconstitution of MCs—whether from $Apoe^{-/-}$ mice or $Apoe^{-/-}Fcer1a^{-/-}$ mice—did not significantly increase recipient mice AAA lesion macrophage content, while lesion T cells, MHC class-II-positive areas, or lesion cell proliferation all tended to recover after BMMCs adoptive transfers, although these trends did not achieve statistical significance (Supplementary Fig S10). These data suggest that IgE actions on MCs have limited impact on macrophage and T-cell infiltration in AAA lesions.

Macrophages in AAA lesions come from blood-borne monocytes, but a short lifespan after repopulation in recipient mice (Leuschner et al, 2012) made it technically difficult to study donor monocytes in AAA recipient mice. Instead, we used macrophages, which do successfully repopulate in recipient mice to study macrophage activation in experimental AAA (Xiong et al, 2009). FACS analysis confirmed that peritoneal macrophages prepared from donor mice contained > 95% macrophages after an adhesion selection to deplete other cells (Supplementary Fig S11). Donor macrophages in AAA lesions from recipient $Apoe^{-/-}Fcer1a^{-/-}$ mice were localized by immunostaining frozen AAA sections with anti-FcεR1a mAb. FcεR1a-positive donor macrophages from $Apoe^{-/-}$ mice appeared in recipient $Apoe^{-/-}Fcer1a^{-/-}$ mouse AAA lesions (Supplementary Fig S8D and Fig 5A, left two panels). Lesions from recipient mice

receiving macrophages from $Apoe^{-/-}Fcer1a^{-/-}$ mice served as negative controls for immunostaining (Supplementary Fig S8D and Fig 5A, right panel). Adoptive transfer of macrophages from either $Apoe^{-/-}$ or $Apoe^{-/-}Fcer1a^{-/-}$ mice did not significantly change plasma IgE levels (Fig 5B), but IgE actions on macrophages contributed to AAA formation. Donor macrophages from $Apoe^{-/-}$ mice, but not those from $Apoe^{-/-}Fcer1a^{-/-}$ mice, increased recipient AAA incidence and post-Ang-II mortality rate. Donor macrophages from $Apoe^{-/-}$ mice, but not those from $Apoe^{-/-}Fcer1a^{-/-}$ mice, also reversed outer aortic diameters in recipient mice, although such change did not reach statistical significance (Fig 5B). Reconstitution of macrophages from $Apoe^{-/-}$ mice, but not those from $Apoe^{-/-}Fcer1a^{-/-}$ mice, also increased lesion macrophage-positive area, T-cell number, and MHC class-II-positive area (Fig 5C), establishing a role of IgE actions on macrophages in AAA lesions.

### Anti-IgE monoclonal antibody administration limits experimental AAA formation

Observations from this study suggest that IgE actions on CD4[+] T cells (Fig 3), MCs (Fig 4), and macrophages (Fig 5) contribute independently to experimental AAAs to different extents. These observations agree with prior studies that CD4[+] T cells, MCs, and macrophages or their activation contribute directly and independently to AAA formation (Xiong et al, 2004, 2009; Sun et al, 2007b). Interference with IgE signaling by genetically depleting the high-affinity IgE receptor FcεR1 (Fig 2), or IgE neutralization using anti-IgE antibodies, therefore could suppress AAA growth. To test this hypothesis, we administered an anti-IgE mAb intravenously 1 day before initiating the AAA (Ang-II mini-pump implantation), and once more at 2 weeks after AAA initiation using doses previously established as effective in allergic mice and humans (Coyle et al, 1996; Haile et al, 1999; Milgrom et al, 1999; Leung et al, 2003; Busse et al, 2011). Two doses of anti-IgE antibody did not change body weight before or after AAA formation (Fig 6A), but significantly reduced plasma IgE by more than 70% (Fig 6B). IgE ablation with anti-IgE antibody reduced AAA incidence, post-Ang-II mortality rate, and outer aortic diameters (Fig 6B), although the differences in AAA incidence and poat-Ang-II mortality rate did not reach statistical significance (Fisher's exact test).

Our leukocyte (CD4[+] T cells, MCs, or macrophages) reconstitution experiments showed that IgE actions on each of these cell types contributed directly, but to different extents, to AAA formation. As we anticipated, anti-IgE antibody treatment, which should block IgE activities on all these target cells, as well as intrinsic vascular cells (Wang et al, 2011), significantly suppressed AAA lesion macrophage content, CD4[+] T-cell number, MHC class-II-positive area, lesion cell apoptosis, microvessel number, and lesion cell proliferation (Fig 6C–F). Non-selective targeting of IgE actions using anti-IgE antibody therapy thus yielded similar effects in suppressing AAA pathogenesis (Fig 6) to those of genetic depletion of IgE receptor FcεR1 (Fig 2).

### IgE actions alter levels of inflammatory cytokines in plasma

One of the most important activities of IgE on inflammatory cells is to induce their pro-inflammatory cytokine production, which may accelerate AAA formation. Reduced AAAs after interruption of IgE

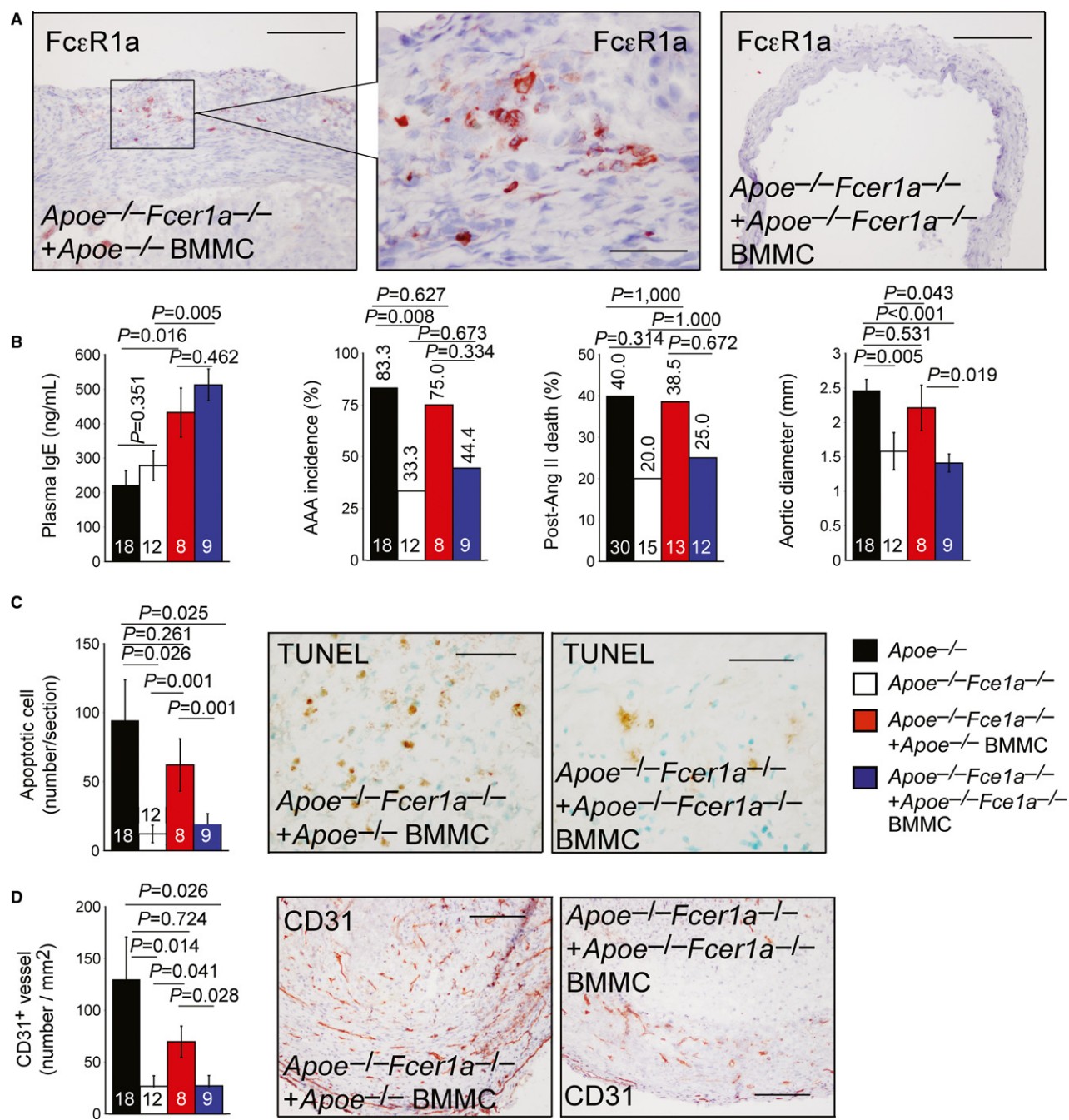

**Figure 4. IgE actions on bone marrow-derived mast cells (BMMCs) in AAAs.**

A   Anti-FcεR1a antibody-mediated immunostaining to detect donor BMMCs from *Apoe*[−/−] mice (left two panels) and *Fcer1a*[−/−]*Apoe*[−/−] mice (right panel) in recipient *Fcer1a*[−/−]*Apoe*[−/−] mice. Scale: 200 μm, insert scale: 50 μm.

B   Plasma IgE level, AAA incidence, post-Ang-II infusion aortic rupture-associated mortality rate, and suprarenal maximal outer aortic diameter.

C, D   (C) AAA lesion TUNEL-positive apoptotic cell percentage; and (D) AAA lesion CD31-positive microvessel number *Apoe*[−/−] mice, *Fcer1a*[−/−]*Apoe*[−/−] mice, and *Fcer1a*[−/−] *Apoe*[−/−] recipient mice receiving donor CD4[+] T cells from *Apoe*[−/−] and *Fcer1a*[−/−]*Apoe*[−/−] mice. Representative AAA lesions from adoptive transferred mice for panels (C) and (D) are shown to the right. Scale: 200 μm.

Data information: (B-D) Data are mean ± SEM. Number of mice per group is indicated in each bar.

interactions with inflammatory cells using either FcεR1-deficient mice or cells, or anti-IgE antibodies, may associate with reduced circulating cytokine levels. In AAA patients, plasma IL6 levels range from 20–40 pg/ml and correlate positively and significantly with indexed aortic diameters ($r = 0.285$, $P = 0.002$, linear regression) (Rohde *et al*, 1999). IL6 deficiency or antibody depletion of plasma IL6 protects mice from aortic elastase perfusion-induced AAAs in mice (Thompson *et al*, 2006), suggesting an important role of IL6 in

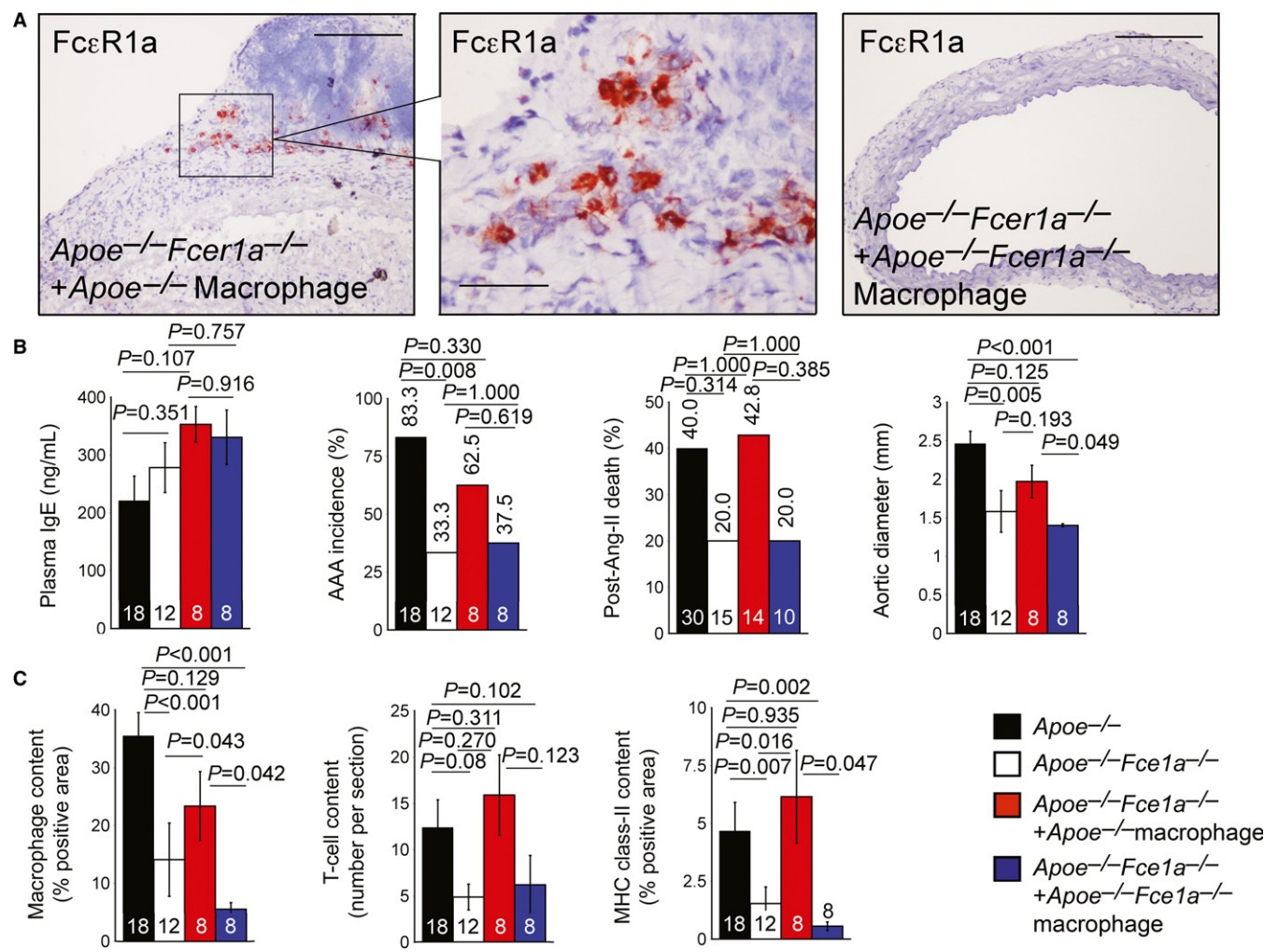

**Figure 5.  IgE actions on macrophages in AAAs.**

A    Anti-FcεR1a antibody-mediated immunostaining to detect donor macrophages from *Apoe*[−/−] mice (left two panels) and *Fcer1a*[−/−]*Apoe*[−/−] mice (right panel) in recipient *Fcer1a*[−/−]*Apoe*[−/−] mice. Scale: 200 μm, insert scale: 50 μm.

B, C  (B) Plasma IgE level, AAA incidence, post-Ang-II infusion aortic rupture-associated mortality rate, and suprarenal maximal outer aortic diameter; and (C). AAA lesion macrophage content, CD4[+] T-cell content, and major histocompatibility complex (MHC) class-II-positive area from *Apoe*[−/−] mice, *Fcer1a*[−/−]*Apoe*[−/−] mice, and *Fcer1a*[−/−]*Apoe*[−/−] recipient mice receiving donor CD4[+] T cells from *Apoe*[−/−] and *Fcer1a*[−/−]*Apoe*[−/−] mice. Data are mean ± SEM. The number of mice per group is indicated in each bar.

AAAs. This study demonstrated that plasma IL6 levels in *Apoe*[−/−] mice resemble those in AAA patients, and fell significantly in *Apoe*[−/−]*Fcer1a*[−/−] mice (Fig 7A). The sensitivity of the ELISA used for IFN-γ did not permit detection and analysis of plasma levels of this important pro-inflammatory cytokine also implicated in AAA pathogenesis (King *et al*, 2009). Low levels of plasma IFN-γ in these mice agreed with the observations that T cells produced 10-fold less IFN-γ than IL6 after IgE stimulation (Fig 1F). Reconstitution of CD4[+] T cells and MCs, but not CD8[+] T cells and macrophages, from *Apoe*[−/−] mice significantly restored plasma IL6 levels (Fig 7B–E). Plasma IL6 levels in recipient mice receiving macrophages from *Apoe*[−/−]*Fcer1a*[−/−] mice were significantly lower than those from recipient mice receiving macrophages from *Apoe*[−/−] mice (Fig 7E). While *Apoe*[−/−] mice receiving IgG treatment showed comparable plasma IL6 levels to those of untreated *Apoe*[−/−] mice, those receiving two doses of an anti-IgE antibody demonstrated significant

reduction in plasma IL6 levels, to the levels from *Apoe*[−/−]*Fcer1a*[−/−] mice (Fig 7F). These observations indicate the prominence of IL6 as a mediator released after IgE activation of inflammatory cells. In contrast, although IgE suppressed IL10 production from isolated CD4[+] T cells *in vitro* (Fig 1E and F), we did not detect significant differences in plasma IL10 levels between *Apoe*[−/−] and *Apoe*[−/−]*Fcer1a*[−/−] mice, or those that received different types of CD4[+] T cells, CD8[+] T cells, MCs, or macrophages (Supplementary Fig S12A–E). Nor did anti-IgE antibody affect plasma IL10 levels in mice with experimental AAAs (Supplementary Fig S12F), although IgE neutralization may affect other untested circulating anti-inflammatory cytokines.

### Increased plasma IgE levels in patients with AAAs

The use of IgE receptor FcεR1-deficient mice, adoptive transfer of various inflammatory cell types, and anti-IgE antibody therapy

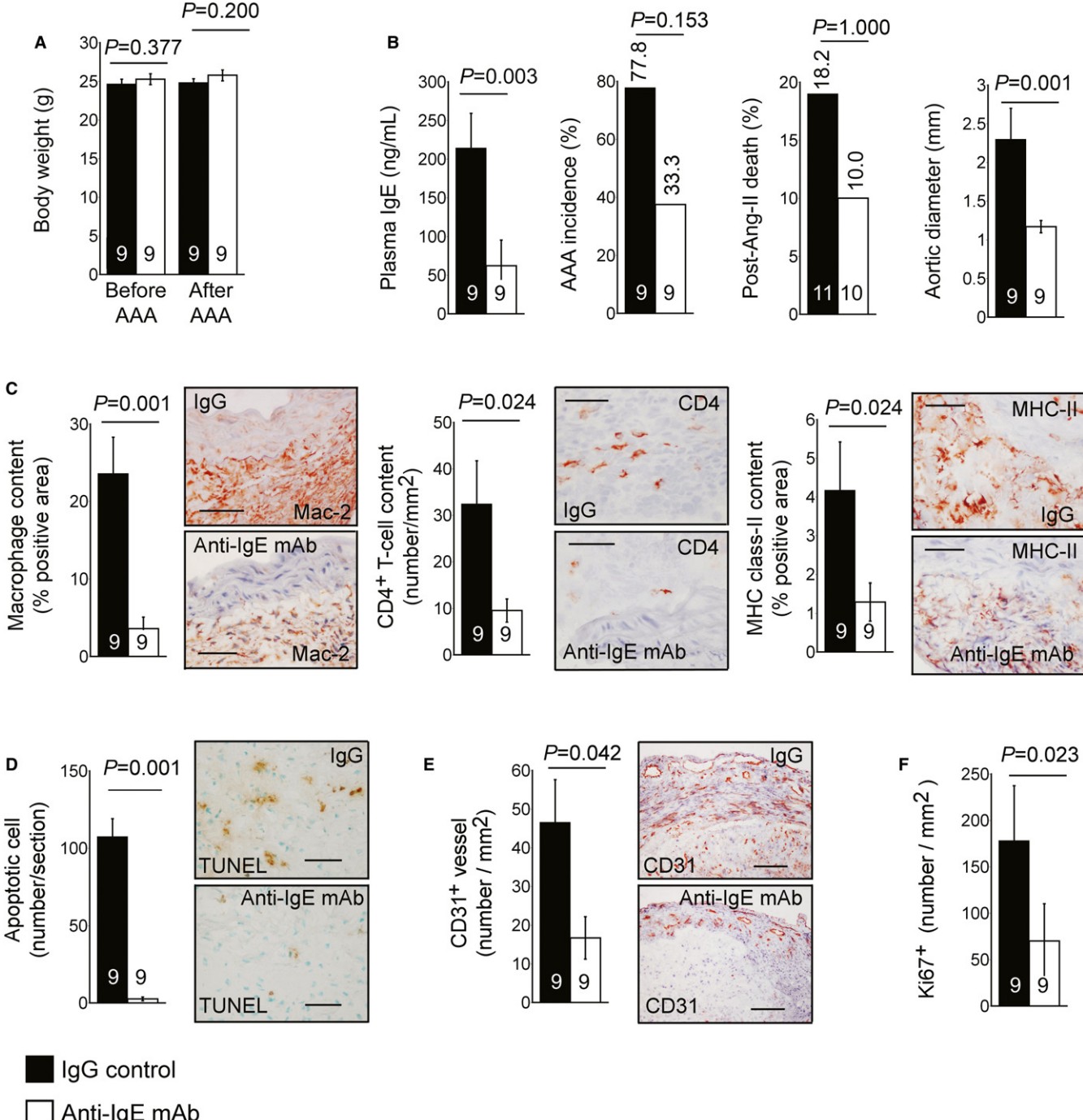

**Figure 6.  Anti-IgE antibody treatment of experimental AAAs.**

A    Body weight in *Apoe*$^{-/-}$ mice that received control IgG and anti-IgE mAb before and after AAA production.

B    Plasma IgE level, AAA incidence, post-Ang-II infusion aortic rupture-associated mortality rate, and suprarenal maximal outer aortic diameter.

C–F  (C) Lesion macrophage content, CD4$^+$ T-cell content, and major histocompatibility complex (MHC) class-II-positive area; (D) Lesion TUNEL-positive apoptotic cell content; (E) CD31-positive microvessel number; and (F) Lesion Ki67-positive proliferating cell number from *Apoe*$^{-/-}$ mice receiving control IgG or anti-IgE mAb.

Data information: Data are mean ± SEM. The number of mice per group is indicated in each bar. Representative figures for panels C to E are shown to the right. Scale: 50 μm.

established that IgE aggravates experimental AAA formation by activating mononuclear leukocytes. Mice have increased plasma IgE under conditions that produce AAAs (Fig 2A and C, left panel). We

have previously reported that AAA patients have significantly higher plasma MC chymase and tryptase levels than controls (Sun *et al*, 2009; Zhang *et al*, 2011). We therefore measured plasma IgE levels

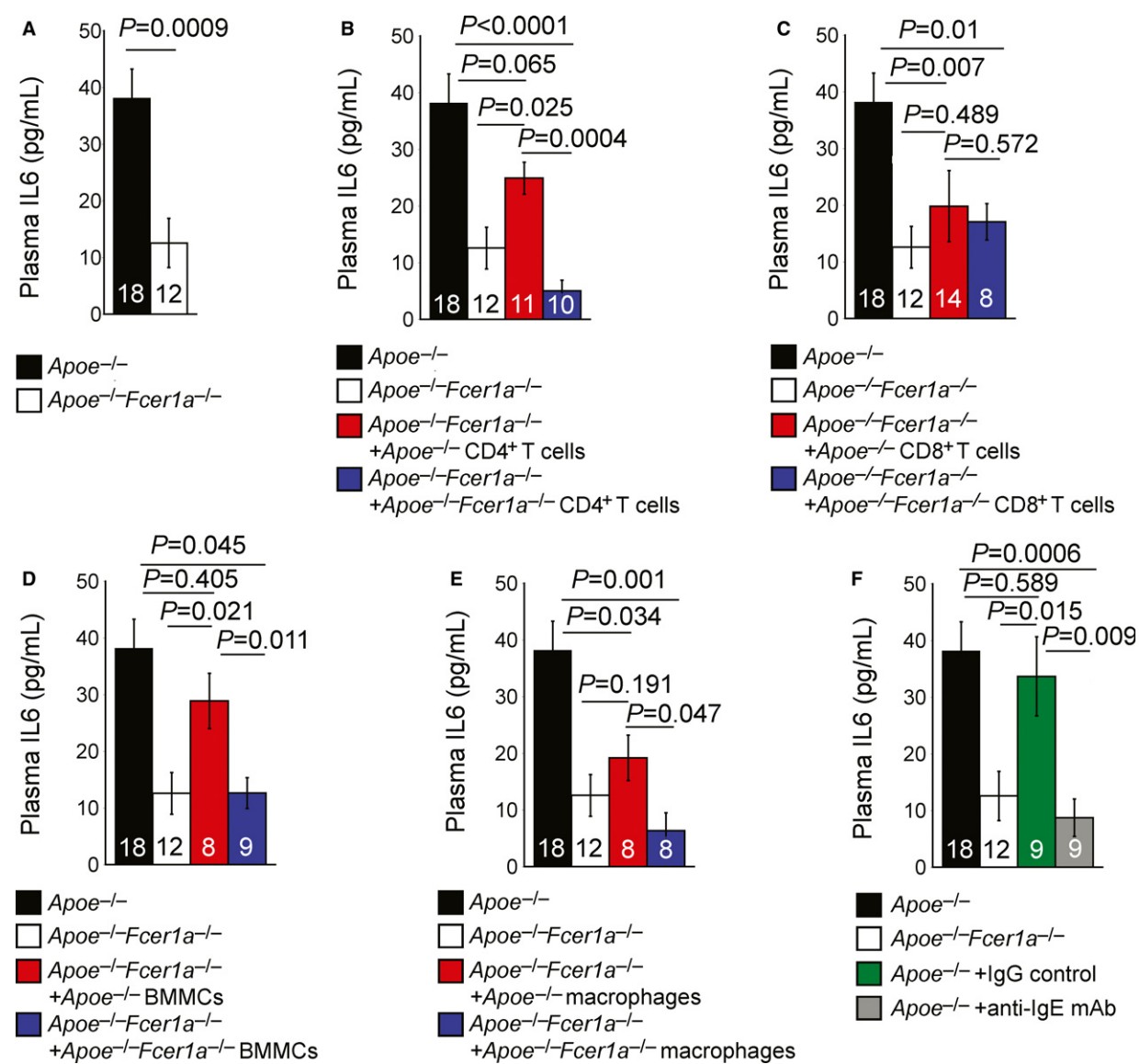

**Figure 7.  Plasma IL6 levels from different groups of mice as determined by ELISA.**

A–F   Plasma IL6 levels in $Apoe^{-/-}$ and $Fcer1a^{-/-}Apoe^{-/-}$ mice (A) and $Fcer1a^{-/-}Apoe^{-/-}$ mice that received adoptive transfer of CD4$^+$ T cells (B), CD8$^+$ T cells (C), bone marrow-derived mast cells (BMMCs) (D), and macrophages (E) from $Apoe^{-/-}$ and $Fcer1a^{-/-}Apoe^{-/-}$ mice, and $Apoe^{-/-}$ mice that received biweekly intravenous administration of anti-IgE mAb or corresponding IgG control (F). Data are mean ± SEM. The number of mice per group is indicated in each bar.

from 487 male AAA patients and compared them by ELISA with those from 200 sex-matched and age-matched individuals without AAAs. Plasma IgE levels showed skewed distribution, therefore data analysis used the non-parametric Mann–Whitney test. AAA patients had significantly higher plasma IgE levels than controls (81.79 ± 61.65 ng/ml versus 7.10 ± 1.70 ng/ml, mean ± SD, $P < 0.001$) (Supplementary Table S1), although we were unable to ascertain whether there were any differences in the events of type I allergies, such as allergic asthma, allergic conjunctivitis, allergic rhinitis, anaphylaxis, angioedema, urticaria, eosinophilia, penicillin allergy, cephalosporin allergy, and food allergy between the two populations because most of these clinical symptoms were diagnosed in community general practices. Receiver–operator characteristic (ROC) curve analyses demonstrated that IgE levels discriminate

between AAA patients and controls (AUC [area under the ROC curve] = 0.588, $P < 0.001$), with optimal sensitivity of 0.60 and specificity of 0.59 (Supplementary Fig S13). Among this population, patients with peripheral arterial disease (PAD) also had significantly higher plasma IgE levels than controls (250.79 ± 229.88 ng/ml versus 15.32 ± 3.33 ng/ml, mean ± SD, $P < 0.001$) (Supplementary Table S1).

**Increased IgE and FcεR1 expression in human AAA lesions**

We have previously shown that IgE activates human macrophages, SMCs, and ECs, and induces their apoptosis (Wang $et\ al$, 2011). The present study suggested a role of IgE stimulation of CD4$^+$ T cells, MCs, and macrophages in experimental AAA formation. Inflammatory

cells in human AAA lesions might also express IgE receptor FcεR1 and bind for IgE. Indeed, CD68-positive macrophages demonstrated expression of FcεR1 and IgE in consecutive frozen sections of human AAAs. Consistent with our prior study (Wang *et al*, 2011), IgE-positive and FcεR1-positive macrophages in human AAA lesions also underwent apoptosis as shown by TUNEL-positivity (Fig 8A). IgE binding to macrophages also triggers cytokine expression from these cells (Wang *et al*, 2011) as in CD4$^+$ T cells. Immunofluorescent staining colocalized the expression of both IgE and FcεR1 on CD4$^+$ T cells in human AAA lesions (Fig 8B).

IgE may also induce SMC apoptosis (Wang *et al*, 2011) in AAA lesions. While regions of human AAA specimens rich in α-actin-positive SMCs lacked detectable IgE signal (Fig 8C), many of the sparse SMCs in other regions bore IgE, observation consistent with a role for IgE in SMC depletion by apoptosis in human AAAs (Fig 8D). Therefore, FcεR1-positive SMCs in AAA lesions may be particularly susceptible to apoptosis (Fig 8E). Some ECs in human AAA lesions also display immunoreactive FcεR1 and IgE (Fig 8F), suggesting that IgE signaling might also contribute to EC death (Wang *et al*, 2011) and intimal erosion, a possibility that merits further investigation.

## Discussion

This study establishes a role for IgE in experimental AAAs. Using Ang-II infusion-induced experimental AAAs in *Apoe*$^{-/-}$ mice, FcεR1-deficient mice, adoptive transfer of leukocyte populations, and anti-IgE mAb administration, we demonstrated that IgE actions on CD4$^+$ T cells, MCs, and macrophages, and possibly other inflammatory and vascular cells contribute to AAA pathogenesis. This study implicates release of pro-inflammatory cytokines after IgE activation of leukocytes a major mechanism for affecting AAAs among other possible pathways.

The present observations show a direct role of IgE in T cells. T-cell functions in AAAs have been controversial, although, it is difficult to compare results obtained from different mouse models of AAA and in different experimental settings. In peri-aortic CaCl$_2$ injury-induced experimental AAAs, absence of CD4$^+$ T cells or Th1 cytokine IFN-γ suppressed AAA formation. Intraperitoneal administration of IFN-γ partially reversed AAA formations in CD4$^+$ T-cell-deficient mice (Xiong *et al*, 2004). In contrast, MHC mismatched aortic transplantation-induced AAAs rely on Th2 cell functions. Allografts from IFN-γ receptor-deficient mice developed severe AAA, which can be blocked by anti-IL4 antibody or compound deficiency of IFN-γ receptor and IL4 (Shimizu *et al*, 2004). Therefore, both Th1 and Th2 cells may contribute to AAA formation differently, depending on the experimental model. In Ang-II infusion-induced AAAs, however, complete elimination of all mature T cells and B cells in Rag-1–deficient mice did not affect AAA incidence or maximal aortic diameters in male or female mice, compared with those with sufficient lymphocytes (Uchida *et al*, 2010). This study by Uchida *et al* may not definitively answer whether T cells or B cells participate in AAAs, but among T cells or B cells, some may promote AAA growth and some may inhibit AAA growth. For example, innate-like B1 cells protect mice from diet-induced atherosclerosis (Sun *et al*, 2010) and may also protect mice from AAA formation, although currently no direct test exists for each T-cell or B-cell

subtype in experimental AAAs. The current study demonstrated FcεR1 expression on CD4$^+$ and CD8$^+$ T cells and a direct role of IgE on T-cell activation. CD4$^+$ T cells expressed much more FcεR1 mRNA and proteins than did CD8$^+$ T cells (Fig 1A, B and D), leading to differences in IL6 and IFN-γ expression in response to IgE stimulation (Fig 1E and F). Our data indicate a role for IL6 from CD4$^+$ T cells in AAAs. Plasma IL6 levels in *Apoe*$^{-/-}$ mice, *Apoe*$^{-/-}$*Fcer1a*$^{-/-}$ mice, and those that received CD4$^+$ and CD8$^+$ T cells from *Apoe*$^{-/-}$ mice and *Apoe*$^{-/-}$*Fcer1a*$^{-/-}$ mice (Fig 7B and C), changed concordantly with AAA lesion sizes (Fig 3C and Supplementary Fig S9C). *In vitro* experiments showed that IgE suppresses CD4$^+$ T-cell IL10 expression (Fig 1E and F), but we did not document significant change in plasma IL10 levels between *Apoe*$^{-/-}$ and *Apoe*$^{-/-}$*Fcer1a*$^{-/-}$ mice, or those that received different types of CD4$^+$ or CD8$^+$ T cells (Supplementary Fig S12A–C). Nonetheless, local level of IL10 in AAA lesions influenced by IgE could occur. The observation of inverse production of IL6 and IL10 in human AAA lesion explant cultures, but not in plasma from AAA patients (Vucevic *et al*, 2012), supports this hypothesis.

IgE activation of MCs leads to MC degranulation and release of inflammatory mediators (Gilfillan & Tkaczyk, 2006; Theoharides *et al*, 2007), including cytokines, chemokines, and proteases. We have previously demonstrated important roles of MCs and MC-derived IL6, IFN-γ, MCP-1, chymases and tryptases in three independent experimental AAA preservations. By releasing these mediators, MCs promote angiogenesis, monocyte migration, and vascular SMC apoptosis (Sun *et al*, 2007b, 2009; Zhang *et al*, 2011, 2012). IgE activation of MCs may enhance the production of these MC mediators and consequent AAA pathogenesis. *Apoe*$^{-/-}$*Fcer1a*$^{-/-}$ mice receiving MCs from *Apoe*$^{-/-}$ mice but not those from *Apoe*$^{-/-}$*Fcer1a*$^{-/-}$ mice have increased AAA lesion apoptosis and microvascularization (Fig 4C and D), consistent with a regulatory role of IgE on MCs. As an indirect mechanism, MCs release IL6 and IFN-γ after activation to induce vascular SMC and EC expression of cysteinyl cathepsins (Sun *et al*, 2007b), which also participate in arterial wall remodeling and AAA pathogenesis (Sun *et al*, 2011, 2012; Qin *et al*, 2012). Indeed, adoptive transfer of cultured MCs from *Apoe*$^{-/-}$ mice into *Apoe*$^{-/-}$*Fcer1a*$^{-/-}$ mice significantly enlarged AAA lesion sizes (Fig 4B) and increased plasma IL6 levels (Fig 7D). The same adoptive transfer of MCs may also change MC-specific proteases, although we are currently unable to quantify mouse plasma chymase and tryptase due to a lack of suitable ELISA kits and to the complexity of different mouse chymase (mouse mast cell protease [mMCP]-1, -2, -4, -5, and -9) and tryptase (mMCP-6, -7, and -11) isoforms.

IgE activation of macrophages leads to macrophage apoptosis and release of IL6, MCP-1, and possibly other pro-inflammatory mediators (Wang *et al*, 2011). To our surprise, adoptive transfer of macrophages from *Apoe*$^{-/-}$ mice did not significantly increase plasma IL6 levels in *Apoe*$^{-/-}$*Fcer1a*$^{-/-}$ mice (Fig 7E). Consistently, outer aortic diameters did not increase significantly in *Apoe*$^{-/-}$*Fcer1a*$^{-/-}$ mice that received macrophages from *Apoe*$^{-/-}$ mice (Fig 5B). These observations reaffirm that IgE regulation of plasma IL6 correlated with AAA growth and also implicate IgE actions on macrophages, including IL6 production, in experimental AAA formation, but for lesser extent than IgE actions on CD4$^+$ T cells or MCs.

IgE stimulation of vascular SMCs and ECs may also affect AAA pathogenesis. We detected negligible IgE in SMC-rich regions in human AAAs (Fig 8C). In contrast, IgE colocalized with SMC

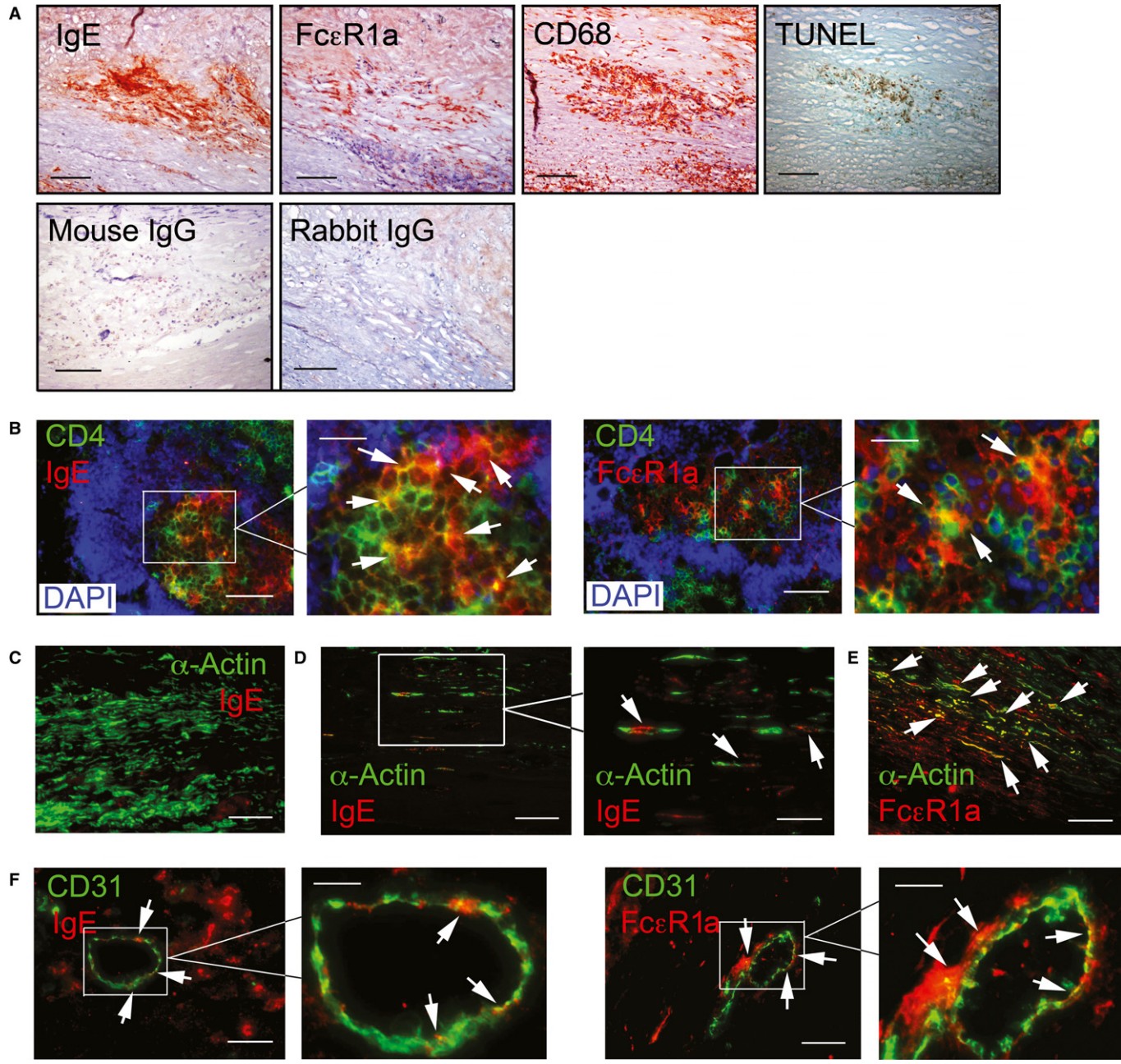

**Figure 8.  IgE and FcεR1 expression in human AAAs lesions.**

A    Immunostaining of human AAA lesion consecutive frozen sections to detect IgE and FcεR1 expression in CD68-positive macrophages and TUNEL reactivity. Mouse IgG and rabbit IgG were used as immunostaining negative controls for mouse anti-IgE and rabbit anti-FcεR1 antibodies.

B    Immunofluorescent double staining of CD4 and IgE or FcεR1 to detect IgE and FcεR1 expression in CD4+ T cells.

C, D    IgE and α-actin immunofluorescent double staining in smooth muscle cells (SMC)-rich and SMC-sparse areas.

E    FcεR1 and α-actin immunofluorescent double staining to detect FcεR1-positive SMCs.

F    CD31 and IgE or FcεR1 immunofluorescent double staining of human AAA lesions.

Data information: Scale: 200 μm; insert scale: 60 μm. Arrows indicate IgE-positive or FcεR1-positive CD4+ T cells, SMCs or endothelial cells (ECs).

paucicellular portions of human AAA lesions (Fig 8D), suggesting a contribution of IgE to SMC apoptosis (Wang *et al*, 2011). In human AAA lesions, IgE also colocalized with ECs (Fig 8F). Our prior *in vitro* experiments from cultured human SMCs and ECs suggested that IgE promotes EC apoptosis and cytokine production (Wang *et al*, 2011). IgE-induced apoptosis of vascular cells may contribute to SMC depletion in human AAA lesions (Fig 8D), and intimal erosion.

In the inflammatory cell reconstitution tests in experimental AAAs, we used CD4+ T cells, CD8+ T cells, MCs, and macrophages

from both $Apoe^{-/-}$ and $Apoe^{-/-}Fcer1a^{-/-}$ mice. We assumed that the only difference between the cells from $Apoe^{-/-}$ mice and those from $Apoe^{-/-}Fcer1a^{-/-}$ mice was the expression of FcεR1. IgE only activates cells from $Apoe^{-/-}$ mice, not those from $Apoe^{-/-}Fcer1a^{-/-}$ mice (Wang *et al*, 2011), but absence of FcεR1 may affect the expression of any other gene product, which may confound our data interpretation from all adoptive transfer experiments (Figs 3, 4, 5 and Supplementary Fig S9). We did not test this possibility in this study. Instead, we treated $Apoe^{-/-}$ mice with anti-IgE antibody while these mice received Ang-II infusion. This antibody should interrupt IgE actions on all cells that express FcεR1. Two doses of anti-IgE antibody to Ang-II-infused $Apoe^{-/-}$ mice effectively ablated plasma IgE and inhibited AAA growth and associated lesion inflammation, apoptosis, microvascularization, and cell proliferation to the same extents as in Ang-II-infused $Apoe^{-/-}Fcer1a^{-/-}$ mice. These observations seem confirmative of IgE functions in AAA from $Apoe^{-/-}Fcer1a^{-/-}$ mice, but have clinical significance. In a similar experiment presented in our earlier study, we used MC-deficient mice to prove a role of MCs in AAAs. In addition, we used over-the-counter MC stabilizers and established an effective prevention of AAA growth in aortic elastase perfusion-induced AAAs (Sun *et al*, 2007b). Anti-IgE antibodies such as omalizumab (also known as rhuMab-E25 or Xolair™) and Talizumab (also known as TNX-901) are currently approved by the U.S. Food and Drug Administration (FDA) for clinical use in patients with allergic asthma, food allergies, or other allergic conditions (Milgrom *et al*, 1999; Leung *et al*, 2003; Busse *et al*, 2011). Effective prevention of experimental AAAs by anti-IgE antibody from this study suggests a potential clinical application of omalizumab and Talizumab among AAA patients.

# Materials and Methods

## Mouse AAA production and lesion characterization

We crossbred $Fcer1a^{-/-}$ mice (C57BL/6, N9, provided by Marie-Helene Jouvin and Jean-Pierre Kinet, Harvard Medical School) with $Apoe^{-/-}$ mice (C57BL/6, N11, The Jackson Laboratory, Bar Harbor, ME) to generate $Apoe^{-/-}Fcer1a^{-/-}$ mice and $Apoe^{-/-}$ control mice. All mice used in this study were littermates in a syngeneic C57BL/6 background. To induce AAAs in $Apoe^{-/-}Fcer1a^{-/-}$ mice ($n = 15$) and $Apoe^{-/-}$ control mice ($n = 30$), anesthetized (200 mg/kg ketamine, 10 mg/kg xylazine, intraperitoneal) 2-month-old male mice were infused with 1000 ng/kg/min Ang-II (Sigma-Aldrich, St. Louis, MO) subcutaneously delivered by Alzet model 2004 osmotic minipumps (DURECT Corp., Cupertino, CA) for 28 days while mice consumed a high-fat diet (C12108; Research Diets, Inc., New Brunswick, NJ). Post-operative analgesia (buprenophine, 0.05 mg/kg/ 12 h, intraperitoneal) was administered every 12 h for 48 h. Mouse body weights were recorded before and after Ang-II infusion. Mouse diastolic and systolic blood pressures and heart rates were determined using the CODA non-invasive blood pressure system (Kent Scientific Co., Torrington, CT). Mice were sacrificed with carbon dioxide narcosis, followed by cardiac puncture blood collection. Plasma IgE, IL6, IFN-γ, IL10, and IgE levels were determined by ELISA according to the manufacturer's protocol (BD Biosciences, San Jose, CA). Plasma total cholesterol, triglyceride, and high-density lipoprotein (HDL) levels were determined using

reagents from Pointe Scientific (Canton, MI). Experimental aneurysms were quantified using the methods of Daugherty *et al* as used in our earlier studies (Daugherty *et al*, 2000; Schulte *et al*, 2010; Zhang *et al*, 2012). The suprarenal maximal aortic diameter of each aneurysm was measured after the peri-adventitial tissue was carefully dissected from the aortic wall. AAA incidence was defined by increase of suprarenal maximal aortic diameter greater than 50% of the mean value from same-age mice that received saline alone ($0.97 \pm 0.02$ mm, $n = 10$), according to previously reported methods (Wang *et al*, 2006). The percentage of AAA incidence and post-Ang-II infusion mortality rate per group were determined. Determination of statistical significances between the groups used the two-tailed Fisher's exact test. All animal procedures conformed with the Guide for the Care and Use of Laboratory Animals published by the U.S. National Institutes of Health and were approved by the Harvard Medical School Standing Committee on Animals (protocol # 03759).

## Mouse aortic tissue immunohistochemical analysis

Aorta segments for immunohistochemistry were cut at the maximal suprarenal outer aortic diameter and embedded vertically with optimal cutting temperature (OCT) compound, and at least 10–15 serial frozen sections covering the maximal dilated aorta were prepared for immunohistochemical analysis as described previously (Schulte *et al*, 2010; Zhang *et al*, 2012). For those with similarly enlarged AAA diameter throughout the thoracic-abdominal aortas, we selected the segment at approximately the same distance from the renal artery as those of others with maximal AAA expansions. In cases with AAA lesions at multiple locations, we selected the largest lesion as close as possible to the same distance from the renal artery as those of others with maximal AAA expansions. Slides of each sample from identical levels were used for staining with each antibody. Serial cryostat cross-sections (6 μm) were used for immunostaining for macrophages (Mac-2, BD Biosciences, 1:1,000), T cells (CD4, 1:90; BD Biosciences; and CD8, 1:300; Abcam, Cambridge, MA), dendritic cells (CD11c, 1:40, BD Biosciences), MHC-II (1:250; BD Biosciences), MCP-1 (1:50, BD Biosciences), elastin (Verhoeff–van Gieson, Sigma-Aldrich), collagen (0.1% Sirius Red F3BA; Polysciences Inc., Warrington, PA), α-actin (SMC, 1:750; Sigma-Aldrich), Ki67 (cell proliferation marker, 1:500; Vector Laboratories, Inc., Burlingame, CA), and CD31 (angiogenesis marker, 1:1,500; BD Biosciences). Lesion apoptotic cells were determined with the *in situ* apoptosis detection kit, according to the manufacturer's instructions (Millipore, Billerica, MA). Elastin degradation and media SMC accumulation were graded according to the grading keys described previously (Sun *et al*, 2007b). T cells, Ki67-positive cells, apoptotic-positive cells, dendritic cells, and CD31-positive microvessels were counted blindly. Unlike aortic cross sections from fixed tissues that allowed counting of most individual immuno-positive signals (Rateri *et al*, 2011), we prepared aortic cross sections from unfixed abdominal aortas optimized for immunohistochemical study, but not for reliable enumeration of all individual immuno-positive cells. We therefore measured macrophage-positive, MHC-II-positive, MCP-1-positive, and collagen-positive areas using computer-assisted image analysis software (Image-Pro Plus; Media Cybernetics, Bethesda, MD). Ang-II-induced AAAs often show regions of disrupted media (Rateri *et al*, 2011), we calculated

AAA lesion areas from regions with both intact and fragmented media (Schulte *et al*, 2010; Zhang *et al*, 2012).

### BMMC, macrophage, T cell culture and adoptive transfer

To prepare BMMCs, bone-marrow cells from *Apoe*$^{-/-}$ and *Apoe*$^{-/-}$ *Fcer1a*$^{-/-}$ mice were cultured for 5–6 weeks in the presence of mouse recombinant IL3 (PeproTech, Inc., Rocky Hill, NJ) and stem cell factor (PeproTech), as reported previously (Sun *et al*, 2007a,b). BMMC purity (nearly 100%) was confirmed with FACS analysis using FITC-conjugated CD117 mAb (eBioscience, Inc., San Diego, CA) (Sun *et al*, 2007a). Mouse thioglycolate-stimulated (3%, Sigma-Aldrich) peritoneal macrophages were harvested from *Apoe*$^{-/-}$ and *Apoe*$^{-/-}$*Fcer1a*$^{-/-}$ mice, cultured in plasma-free medium for 2 h, suspension cells removed, adhesive macrophages harvested, and macrophage purity examined using FACS with anti-CD11b-FITC and anti-F4/80-APC mAbs (Caltag Laboratories, Burlingame, CA). Peripheral CD4$^+$ and CD8$^+$ T cells from splenocytes of 8-week-old *Apoe*$^{-/-}$ and *Apoe*$^{-/-}$*Fcer1a*$^{-/-}$ mice were first purified using a Nylon column according to the manufacturer's instructions (Polysciences, Inc., Warrington, PA), followed by antibody/complement-mediated depletion of B cells and CD8$^+$ T cells (purified I-A$^b$ and CD8a antibodies, BD Pharmingen) or B cells and CD4$^+$ T cells (purified I-A$^b$ and CD4 antibodies, BD Pharmingen). Purified CD4$^+$ and CD8$^+$ T cells were further purified by MACS-sorting using mouse T-cell isolation kits according to the manufacturer's instructions (Miltenyi Biotec Inc., Auburn, CA). Cell purity was confirmed by FACS analysis using anti-CD3-APC, anti-CD4-PE, and anti-CD8-Alex700 mAbs (all from eBioscience, Inc.).

To reconstitute cells *in vivo*, 8-week-old male *Apoe*$^{-/-}$*Fcer1a*$^{-/-}$ mice were given *in vitro* prepared BMMCs (*n* = 18 for *Apoe*$^{-/-}$ cells and *n* = 12 for *Apoe*$^{-/-}$*Fcer1a*$^{-/-}$ cells), peritoneal macrophages (*n* = 14 for *Apoe*$^{-/-}$ cells and *n* = 10 for *Apoe*$^{-/-}$*Fcer1a*$^{-/-}$ cells), CD4$^+$ (*n* = 20 for *Apoe*$^{-/-}$ cells and *n* = 15 for *Apoe*$^{-/-}$*Fcer1a*$^{-/-}$ cells), and CD8$^+$ (*n* = 25 for *Apoe*$^{-/-}$ cells and *n* = 10 for *Apoe*$^{-/-}$ *Fcer1a*$^{-/-}$ cells) T cells by tail vein injection (1 × 10$^7$ cells per mouse). Unlike other methods of bone-marrow-cell transplantation, irradiating the *Apoe*$^{-/-}$*Fcer1a*$^{-/-}$ recipient mice before receiving donor cell intravenous transfer is not necessary, and donor cells migrate to most organs and injured aortas. Recipient mice were introduced to the Ang-II infusion-induced experimental AAAs 1 day after tail vein injection. Donor cells in AAA lesions were confirmed with immunostaining aortic sections with hamster anti-mouse FcεR1a mAb (1:100, eBioscience, Inc.) at harvesting. To minimize inter-experimental variations, we used the Ang-II and minipump from the same lot number and had the same investigator perform implantation. We also planned the procedures on various experimental groups including *Apoe*$^{-/-}$ and *Apoe*$^{-/-}$*Fcer1a*$^{-/-}$ control mice so that some mice in each group were done each day, rather than having only one condition processed per day.

### Anti-IgE antibody treatment

Two-month-old male *Apoe*$^{-/-}$ mice received tail-vein injection of rat anti-mouse IgE mAb (*n* = 11) in a dose previously validated in mice (Coyle *et al*, 1996; Haile *et al*, 1999) (330 μg in 200 μl of saline per 25 g body weight, BD Pharmingen) 1 day before the surgery. Matched rat IgG1 isotype (*n* = 10, BD Pharmingen) was used as

negative control. Mice received a second dose of the same antibody or IgG1 isotype 14 days after surgery. Mice were harvested 28 days after initial Ang-II infusion.

T-cell real-time polymerase chain reaction (RT-PCR), FACS, immunoblot analysis CD4$^+$ and CD8$^+$ T cells (2.5 × 10$^6$/ml) were cultured in a complete medium (RPMI 1640 medium and 10% fetal bovine serum) in anti-CD3 (1 μg/ml) mAb (BD Pharmingen) pre-coated culture dishes. After treatment with different stimuli, including INF-γ (20 ng/ml), TNF-α (10 ng/ml), IL6 (20 ng/ml), IgE (50 μg/ml), and Ang-II (100 nM), total cellular RNA was extracted using Qiagen RNA isolation kit. Equal amounts of RNA were reverse transcribed, and quantitative PCR was performed in a single-color RT-PCR detection system (Stratagene, La Jolla, CA). The mRNA levels of FcεR1-α, FcεR1-β, and FcεR1-γ chains were normalized to those of β-actin.

For flow cytometry, cells were stained with the proper combination of antibodies and analyzed on a flow cytometer FC500 (Beckman Coulter, Brea, CA). The following antibodies were used for T-cell flow cytometry: anti-CD3-APC, anti-CD4-PE, anti-FcεR1a-biotin, anti-CD8-Alex700 mAbs (all from eBioscience, Inc.). For surface staining, cells were incubated with antibodies for 20 min at 4°C. For intracellar staining, cells were fixed and permeabilized according to the manufacturer's instructions (eBioscience, Inc.) before adding the anti-FcεR1a-biotin antibody. Isotype controls were used to correct compensation and to confirm antibody specificity. All samples were analyzed using flow cytometry on a FACSCalibur (BD Biosciences).

For immunoblot analysis, an equal amount of protein from each cell type preparation was separated by SDS–PAGE, blotted, and detected with mAbs against mouse FcεR1α (1:1,000; eBioscience, Inc.) and β-actin (used for protein loading control, 1:1,000; Santa Cruz Biotechnology Inc., Santa Cruz, CA).

### T-cell survival, proliferation, and cytokine production

To test whether IgE affects T-cell survival and proliferation, we seeded different numbers of CD4$^+$ or CD8$^+$ T cells (2.5 × 10$^5$ to 1 × 10$^6$ per well) on a 96-well flat-bottomed tissue culture plate containing both anti-CD3 (1 μg/ml) and anti-CD28 (1 μg/ml) antibodies. Cells were treated with murine IgE (50 μg/ml; Sigma-Aldrich) for 3 days. MTT assay was then used to detect cell survival, according to the manufacturer's instructions (Millipore). Medium absorbance was read at 570 nm. ELISA (BD Biosciences) was then used to determine IL12 levels from T-cell culture media.

To detect cytokine production in CD4$^+$ and CD8$^+$ T cells, T cells (2.5 × 10$^6$/ml) were cultured in a complete medium (RPMI 1640 medium and 10% fetal bovine serum) in anti-CD3 mAb (1 μg/ml, BD Pharmingen) pre-coated plates in the presence or absence of different doses of IgE (0–100 μg/ml; Sigma-Aldrich) for 2 days. Cells were collected for total RNA preparation using the Qiagen RNA isolation kit for RT-PCR (Bio-Rad, Hercules, CA) to determine mRNA levels of IL6, IFN-γ, and IL10. Cell culture media were collected to measure secreted cytokines by ELISA, according to the manufacturer's instructions (BD Biosciences).

### Patient population and IgE ELISA

In an ongoing randomized population-based screening trial for AAAs, PAD, and hypertension in more than 50,000 men 65–

74 years of age in the mid-region of Denmark (Grøndal *et al*, 2010), baseline plasma samples were taken consecutively at diagnosis of 476 AAA patients and 200 age-matched controls without AAA or PAD. AAA was defined as having maximal aortic diameter greater than 30 mm, and PAD was defined as an ankle-brachial index (ABI) lower than 0.90 or >1.4. Ankle systolic blood pressure was measured as previously validated and reported (Joensen *et al*, 2008), and maximal anterior–posterior diameter of the infrarenal aorta was measured in the peak of the systole from inner edge to inner edge of the aorta. Patients with AAAs less than 50 mm were offered annual control scans by the screening team; patients with AAAs measuring 50 mm or larger were referred for a computed tomography (CT) scan and vascular surgical evaluation. The inter-observer variation of aortic diameter measurements was 1.52 mm (Grøndal *et al*, 2012). Growth rates of small AAAs in patients kept under surveillance were calculated by individual linear regression analysis, using all observations. Blood samples were centrifuged at 3,000 g for 12 min, aliquoted, and stored at −80°C until analysis was performed. All subjects gave informed consent before partici-pating, and the Local Ethics Committee of the Viborg Hospital, Denmark, approved the study, which was performed in accordance with the Helsinki Declaration. The Partners Human Research Committee (Boston, MA) also approved the use of non-coded human samples. Plasma IgE levels were determined using the Human IgE Ready-SET-Go! according to the manufacturer's instruc-tions (eBioscience, Inc.).

### Human AAA lesion immunohistology

Immunohistochemistry with mouse anti-human IgE (1:70, Southern Biotech, Birmingham, AL) and rabbit anti-human FcεR1 (1:300; Santa Cruz Biotechnology) and their colocalization with apoptosis (TUNEL, Apoptosis Tag kit; Millipore) was performed on acetone fixed frozen sections from human AAA ($n = 11$) and control normal aortas ($n = 5$) by avidin/biotin method. Specimens obtained by protocols approved by the Human Investigation Review Committee at the Brigham and Women's Hospital were immediately immersed in saline, embedded in OCT, and stored at −80°C freezer. To local-ize IgE and FcεR1 to vascular SMCs, ECs, and CD4$^+$ T cells, we performed double immunofluorescent staining ($n = 11$) with mouse anti-IgE (1:10) or rabbit anti-FcεR1 (1:80) antibodies incubated over-night, followed by anti-mouse or anti-rabbit Alexa-555 (Invitrogen, Grand Island, NY). Sections were washed, blocked with 4% of appropriate normal serum, and incubated with mouse anti-α-actin (1:40, HHF35; Enzo-Diagnostics, Syosset, NY), CD31 (1:40, Dako, Carpinteria, CA), and CD4 (1:20, Dako), followed by anti-mouse Alexa-488 (Invitrogen).

### Statistical analysis

One sample Kolmogorov–Smirnov test and probability plot (not shown) were used to determine whether human plasma IgE levels were normally distributed, and compared between controls and cases with Student's *t*-test and the Mann–Whitney *U* test. ROC curve analyses were performed non-parametrically to test the predictive value of the test, concerning the prediction of AAA cases. For analy-ses of the ROC curves, the null hypothesis was that the test performed similarly to the diagonal line—i.e. the area under the

### The paper explained

#### Problem

We previously showed that immunoglobulin E (IgE) activates cultured MCs and macrophages, both are important pro-inflammatory cells in human and experimental AAAs. However, it remains unknown whether IgE activation of these cells or other inflammatory cells contributes to AAA pathogenesis, and whether it is possible that inter-ruption of this IgE activity ameliorates AAA development.

#### Results

This study demonstrates that not only MCs and macrophages, but also CD4$^+$ and CD8$^+$ T cells express IgE receptor FcεR1. IgE induces CD4$^+$ T-cell production of pro-inflammatory cytokines IL6 and IFN-γ, but reduces their production of anti-inflammatory cytokine IL10. Genetic depletion of this IgE receptor protects mice from chronic angiotensin II infusion-induced AAAs. We are also able to reverse reduced AAAs and plasma IL6 levels in FcεR1-deficient mice by repopu-lating CD4$^+$ T cells, MCs, and macrophages from FcεR1-sufficient mice but not those from FcεR1-deficient mice. Relevant to clinical implica-tion, we demonstrate that biweekly intravenous administration of an anti-IgE monoclonal antibody reduces plasma IgE by more than 70% and yields similar suppression of AAAs to those in FcεR1-deficient mice. In humans, we also discover that AAA patients have higher plasma IgE levels than those without AAAs.

#### Impact

This study establishes an important role of IgE in AAA formation by activating inflammatory cells, including CD4$^+$ T cells, MCs, macro-phages, and possibly other untested cell types. Reduced AAAs in mice that are genetic deficient for IgE receptor FcεR1 or that receive treat-ment with an anti-IgE monoclonal antibody suggest a future therapeu-tic potential of this aortic disease using anti-IgE antibodies, such as omalizumab and Talizumab that are currently used widely among patients with asthma or other allergic diseases.

curve was 0.5. If the lowest 95% confidence limit for the area under the curve was above 0.5, a significant predictive test was present. The optimal cut-off points were determined, and the respective sensi-tivity and specificity were calculated. The associations of plasma IgE were correlated to maximal aortic diameter, lowest ABI, and AAA growth rate by Spearman's correlation test. Because of relatively small sample sizes and often skewed data distribution, we selected the non-parametric Mann–Whitney *U* test for paired data sets and one-way ANOVA with post-hoc Bonferroni test was used for compar-ison among three or more groups to examine statistical significance for all data from cultured cells and experimental AAAs. Fisher's exact test was used to compare the differences in AAA incidence and post-Ang-II mortality. $P < 0.05$ was considered statistically significant.

**Supplementary information** for this article is available online: http://embomolmed.embopress.org

### Acknowledgements

The authors thank Mrs. Eugenia Shvartz for technical assistance, Ms. Sara Karwacki for editorial assistance, and Drs. Marie-Helene Jouvin and Jean-Pierre Kinet of Beth Israel Deaconess Medical Center and Harvard Medical School for providing the FcεR1α-deficient mice. This study is supported by grants from the National Institutes of Health (HL60942, HL81090, HL88547 to GPS; HL34636, HL80472 to PL), and by an American Heart Association Established Investigator Award (0840118N, to GPS).

## Author contributions

JW, GKS, MAS, HC, AH, and YW performed the mouse experiments and human sample analysis. JSL, MX, and JAW collected human specimens and data analysis. MX, and NX helped with FACS analysis. PL helped with manuscript writing and data explanation. GPS designed the experiments and wrote the manuscript.

## Conflict of interest

The authors declare that they have no conflict of interest.

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
