## [Review Process File · EMBO Molecular Medicine]

IgE actions on CD4⁺ T cells, mast cells, and macrophages participate in the pathogenesis of experimental abdominal aortic aneurysms

Jing Wang, Jes S. Lindholt, Galina K. Sukhova, Michael A. Shi, Mingcan Xia, Han Chen, Meixiang Xiang, Aina He, Yi Wang, Na Xiong, Peter Libby, Jian-An Wang, Guo-Ping Shi

Corresponding author: Guo-Ping Shi, Brigham and Women's Hospital

Review timeline:

Submission date:	31 December 2013
Editorial Decision:	13 February 2014
Revision received:	29 April 2014
Editorial Decision:	21 May 2014
Revision received:	21 May 2014
Accepted:	22 May 2014

Transaction Report:

Editor: Céline Carret

1st Editorial Decision

13 February 2014

Thank you for the submission of your manuscript to EMBO Molecular Medicine. We have now heard back from the three referees whom we asked to evaluate your manuscript. As you will see from the reports below, the referees find the topic of your study of potential interest. However, they do raise substantial concerns on your work, which should be convincingly addressed in a major revision of the present manuscript.

As I said, although the referees find the study to be important and novel, they also raise a number of issues about the conclusiveness of the results and several technical issues. Referee 1 feels that the study needs to be completely re-written and offers very good suggestions to improve the article conclusiveness and clarity. Referee 2 is concerned by the model selected and along with the other two referees recommend new experimental display and controls. As for referee 3, while appreciating the study, this referee is also concerned by the limited causative evidence provided.

Overall it is clear that publication of the manuscript cannot be considered at this stage. I also note that addressing the reviewers concerns in full will be necessary for further considering it in our journal and this appears to require a lot of additional work, experimentation and time. I am unsure whether you will be able or willing to address those and return a revised manuscript within the 3 months deadline. On the other hand, given the potential interest of the findings, I would be willing to consider a revised manuscript with the understanding that the referees' concerns must be fully

addressed and that acceptance of the manuscript would entail a second round of review.

I should remind you that it is EMBO Molecular Medicine policy to allow a single round of revision only and that, therefore, acceptance or rejection of the manuscript will depend on the completeness of your responses included in the next, final version of the manuscript. For this reason, and to save you from any frustrations in the end I would strongly advise against returning an incomplete revision and would also understand your decision if you choose to rather seek rapid publication elsewhere at this stage.

I look forward to seeing a revised form of your manuscript as soon as possible.

Should you find that the requested revisions are not feasible within the constraints outlined here and require additional time to complete the revision or choose to submit your paper elsewhere, we would welcome a message to this effect.

***** Reviewer's comments *****

Referee #1 (Comments on Novelty/Model System):

This is an interesting article detailing the potential importance of IgE and its high affinity receptor in abdominal aortic aneurysms (AAA). The paper appears to have significant novelty, but suffers greatly due to the way in which the data are presented. It essentially needs a complete overhaul in the writing. Despite this fact, I feel that the paper does have significant potential and clinical relevance. A revised version might indeed be an important paper and therefore should be considered for publication. I will detail these issues on a point by point basis in my remarks to the author.

Referee #1 (Remarks):

I was quite impressed by the novelty of the work presented in this manuscript. The idea that IgE and its high affinity receptor contribute to AAA is important, especially given the potential for treatment using anti-IgE. However, the paper in its current form is a very difficult read. Without exaggeration, it took me more than twice as long to read this article as it likely should have. Throughout the paper, important aspects of many figures are unclear, methods are not explained very well, and often the text states that there are important differences that are not supported by the statistics shown in the figures. At times I was all but baffled. The paper needs significant rewriting. I will detail my suggestions below.

1. Figure 1A begins with a logical assessment of IgE receptor gene expression using qPCR. These data are somewhat difficult to interpret as we don't have a comparison sample that is known to express these genes. Can the authors show similar values for a mast cell or basophil? Similarly, part B shows very low expression of the receptor on the surface of CD4 T cells. It is important for the reader to know if the background staining with IgG has been subtracted or not. It is also quite confusing that CD8 T cells express similar amounts of this receptor while having no detectable mRNA. I was also surprised that T cell activation with IgE was carried out in the absence of antigen. This seems to be a glaring omission, since the receptor signals far stronger when aggregated by antigen. The use of antigen in figures 1D and 1E is needed. Finally, in figure 1E the concentration of cytokines is stunningly large, especially for IL-10. Are the authors certain that these units are correct?

2. The details of the in vivo model employed throughout the paper are first presented in figure 2. These data were somewhat difficult to interpret, as it was not clear at what point measurements were made. Are all of the animals analyzed on day 28, or are there animals that die of aneurysms prior to day 28 - and are they analyzed at the point of death? This is an important detail, because if the animals that die of aneurysms prior to day 28 are not included in the data analysis, this is a critical omission that could greatly skew the results. It certainly appears that many animals die prior to day 28, as the number of animals employed in the assays is significantly higher when assessing death compared to all other measurements shown in figure 2.

3. Figure 3,4 and 5 offer a logical progression in which the authors investigate the importance of CD4 T cells, mast cells, and macrophages in AAA. They also determine the importance of IgE receptor expression on each lineage. I thought this was a very strong mechanistic approach to this interesting *in vivo* model. It was unfortunate that these figures suffered from a combination of confusion in the way that some data are presented, and confounding statistics that do not agree with what is stated in the text. Because these problems are fairly consistent throughout the figures, I will point them out simultaneously. First, the use of staining for the high affinity IgE receptor to determine the relative engraftment success of the injected cells has some strength and logic to it. However, this approach suffers in that it does not allow one to determine if the receptor-negative cells were similarly engrafted. This is of course a very important point, sense of failure of these latter cells to engraft would lead to the kind of negative data that these figures show. Can they use CFSE or some other means to show proper engraftment of FcεRI-negative cells? Secondly, the way in which plasma IgE levels are displayed is quite confusing. In figure 2B, it appears that we are supposed to compare the bars in the 1st 2 graphs to each other. I say this because the authors state that there was a general increase in IgE levels after T cell transfer. However, the bars are on different graphs and there is no statistical comparison of the red and blue bars to black and white bars. So it was not clear to me how one can state that IgE levels were increased by the cells. More importantly, the relative impact of cell engraftment on aneurysm incidence and death rate is often stated to be significant in the text when discussing figures 3,4 and 5. However, the statistical analyses shown in these figures clearly does not support these statements. I did not understand why the authors would make a claim of statistical significance and then show P values so clearly above .05. I feel that I must have misread the paper, but I read the sections over and over again. Finally, these figures also employed a histological assessment of macrophage content that is displayed as percent positive area. While I do not consider myself an expert in histology, this means of displaying the data strikes me as unreliable. I would be happy to hear the authors explanation for this. But in my own interpretation, it would appear that a great many more macrophages could be present in a small area and yet would be counted as less inflammation due to the fact that they occupy a smaller area of tissue. It seems that the authors should instead enumerate macrophages as they did with T cells.

4. The data shown in figure 6 employing anti-IgE therapy have tremendous potential and the authors should be complimented for performing these experiments. The main issue with this figure is portion B, where the authors once again state that incidence and death rates from aneurysms are diminished by the therapy, when the P values do not support this.

5. In figure 7, the authors analyze the impact of different cell transfers or anti-IgE therapy on interleukin 6 production. This is a great idea, but there were 2 aspects of this figure that were confusing. First, I expected a statistical comparison between recipient mice receiving T cells, mast cells, or macrophages and their respective control groups not receiving cells (part a). This is a simple issue that could just be mentioned in the text. The 2nd potential problem with this figure was simply that the amount of IL-6 measured in the plasma is so vanishingly small. I don't really know what the authors can do about this, but at 20 ng/ml, these measurements are likely at the very bottom of their assay, and it is not immediately clear to the reader that this amount of cytokine has any meaning to the disease.

6. The final figure in the paper has tremendous potential, similar to figure 6. It suffers from the way it is presented, however. I left this figure not knowing exactly what it really showed - which is disappointing because I think the data may be quite important. It would seem that if the authors wish to demonstrate IgE and IgE receptor expression co-localizing with specific cell types in the aneurysm tissue, they really need to do confocal microscopy and present it in a way that offers some sort of statistical relevance when performed on samples from multiple patients. In its current form, we are only shown one representative panel from a single patient. The anti-IgE staining is done at a very high concentration (1:10) and hence needs an IgG control to be convincing. Part A does not use confocal microscopy, and when confocal microscopy is use, it is not clear in most of the figures that any yellow staining indicating co-localization is present (with the exception of figured E). I think the authors really need to rethink how they present these potentially important data.

On balance, this was a novel and interesting paper that was extremely difficult to read and often made statements not supported by statistics. I think it has real potential but needs an entire overhaul in the writing.

Referee #2 (Comments on Novelty/Model System):

Setup of the T cell adoptive transfer (single injection, before AAA induction) and antibody blocking experiments (2 injections over a 4 weeks period) raise some questions (see below review)

Referee #2 (Remarks):

This is a very relevant study, identifying IgE as a major actor in AAA pathogenesis. Nevertheless, I have some concerns, mainly pertaining the experimental setup of the study.

Major points:

1) anti-IgE treatment 2 administrations, over 28 days. Given the expected plasma half-life of IgG (1-2 days), this suggests only partly coverage of the complete AAA development time span. For a better appreciation of the effects, the authors may show effective IgE blockage capacity in plasma at 1 and 2 weeks after injection.

2) I am slightly concerned regarding the adoptive transfer scheme. CD4/8 cells were transferred only once, 1 day before start of treatment. It's remarkable that this has such dramatic effects. Where do these transferred cells reside; what is their turnover, and do transferred cells undergo clonal expansion during AAA (dynamics data on CD45.1->CD45.2 transfer in AAA would be insightful)

3) The statistical analysis applied for the adoptive transfer data is not very meaningful, and does not allow conclusions regarding the overall efficacy of intervention (for this a direct comparison with AAA in WT vs FcER1ko is required). This is particularly worrisome as in our hands AAA models display considerable inter-experimental variation in lesion development; and as data presentation suggests that adoptive transfer involve separate experiments (and thus seem to lack the proper controls (untreated-plain FcER1ko))

4) To my surprise, FcER1ko mice show similar levels of IgE as WT controls (Fig 3), despite the absence of AAA lesions in 67% of mice; do IgE levels correlate with severity of disease? FcER1ko IgE data should preferably be plotted as done for WT mice (Fig 2a) shown in the same way as s. Differentiate Fig 2a Data should be presented as If not (which the presented data suggest) how do authors explain this seeming paradox?

Minor comments:

- 1) how does T cell FcER1 expression compare to that on MC and mf (FACS -MFI or qPCR). Panel 1b: are FcER1+ populations separated at baseline; please show typical FACS plot, and MFI for FcER1 positive and negative populations)
- 2) Are plasma IgG and IgM levels also changed in AAA
- 3) What is the relative abundance of FcER1+ MC, macrophages and T cells in AAA lesions (and control arteries)? Please quantify double stainings (or perform aorta FACS)?
- 4) BMDC transfer studies: please show lesion T cell and macrophage content data.
- 5) Are plasma or lesion IL17 levels upregulated?

Referee #3 (Remarks):

Interesting and important work

Major issues

The authors claim that T cells express FcεRI, based on several suggestive but circumstantial lines of evidence. The authors should show FACS results for FcεRI expression by T cells. How does T cell fceRI expression compare to that of mast cells, basophils, etc? What is different about FcεRI T cells? Can we be sure that the FcεRI positive cells are T cells vs contaminating cells. What were the contaminating cells? The authors should also discuss the existing and published evidence for FcεRI expression by T cells.

The authors conclude that IgE action on FcεRI contribute to the phenotype described. How IgE do that? Is cross linking of FcεRI involved or required? What are the characteristics of the IgE that was used to stimulate T cells and of the IgE that was detected in vivo? High cytokinergic? Antigen specific?

The authors conclude from their adoptive transfer experiments that mast cells, T cells and macrophages are important for AAA. The information on the methods employed is scarce, but it appears that AAA formation was induced one day after the adoptive transfer. Do the authors think that repopulation of tissues by adoptively transferred cells has occurred? Have the authors checked for that (other than by fceRI staining? Have the authors employed double staining protocols? If so, what were the results?

It remains unclear if/how FcεRI+ T cells affect AAA formation. The authors claim that "CD4+ T cells from Apoe^{-/-} mice partially or fully restored both the AAA incidence rate and post-Ang-II mortality rate in Apoe^{-/-}FcεRI1a^{-/-} recipient mice much greater than those from Apoe^{-/-}FcεRI1a^{-/-} mice", but the results shown do not back this claim, as evidenced by the lack of significance of the results shown in fig 3.

This criticism also applies to the conclusions drawn from the macrophage transfer experiments. The view that "CD4+ T cells from Apoe^{-/-} mice partially or fully restored both the AAA incidence rate and post-Ang-II mortality rate in Apoe^{-/-}FcεRI1a^{-/-} recipient mice much greater than those from Apoe^{-/-}FcεRI1a^{-/-} mice" does not match the lack of significance (Fig 5).

Minor issues

The authors state that "IgE ablation with anti-IgE antibody reduced AAA incidence and post-Ang-II mortality rate" but fail to mention that this effect was not statistically significant.

"AAA patients had significantly higher plasma IgE levels than controls". were the two populations checked for confounding factors such as type I allergies?

I do not detect any double double positive cells in the CD4/IgE CD4/FcεRI stains shown in fig 8.

1st Revision - authors' response

29 April 2014

Referee #1 (Remarks):

A case is made at the level of plausibility for an association of a TASK4 missense mutation with the PCCD/IVF phenotype in a single 63-year old patient. The TASK4 mutation confers gain of function of a plateau-type K current in oocytes. Together with a splice site variant in SCN5a which is predicted to cause loss of function, the authors argue that the genotype may explain the phenotype.

The case is plausible. It would have been further tested had any relatives been phenotyped and genotyped, but no family history is mentioned. Such information would substantially strengthen the manuscript.

Thank you for the positive comments and for reviewing our manuscript. We absolutely agree that having a genotype-phenotype correlation for the family would substantially strengthen the manuscript and support our experimental conclusions. We tried hard to convince other family members, in particular first degree relatives to participate in our study. However, they clearly and strictly refused to join. Still, we were able to reconstruct the family history, at least, and already introduced this into the original manuscript (page 7). Here we previously noted *"The family history was negative for sudden cardiac death or known inherited cardiac conditions"*. In addition, we now explicitly state on page 10: *"Since DNA from other family members was not available, we were not able to proof whether the identified genetic mutations in both genes were inherited or occurred as de-novo ones, however the family history was not further indicative for other arrhythmias or sudden cardiac death"*.

Nevertheless, our study is as we think, an excellent example that the novel technique of whole exome sequencing, combined with PPT predictions and electrophysiological recordings, can provide answers in single arrhythmia cases, where in former days classical genetics would have failed.

Referee #2 (Remarks):

In this manuscript, the authors applied whole exome sequencing (WES) with a prioritization algorithm for recognizing disease-causing mutations, to identify a new channel mutation that appears to contribute to a severe cardiac abnormality in a human patient (PCCD: progressive cardiac conduction disorder; and IVF: idiopathic ventricular fibrillation). Thus, in addition to a mutation in the SCN5A Na channel gene, they also find a glycine-to-arginine substitution in KCNK17 that encodes the TASK-4 background K channel. Expression of TASK-4 was prominent in human cardiac conduction tissue and, in heterologous expression studies, the G88A mutation caused a strong gain-of-function in K current. Based on this evidence, the authors conclude that these mutations together conspire to yield the pronounced cardiac phenotype in this patient.

Overall, this a well-written and well-illustrated paper that provides new information supporting the idea that TASK-4 mutations can contribute to human arrhythmias. It also suggests a previously unappreciated role for TASK-4 in cardiac conduction systems. Finally, and also importantly, it presents a rational blueprint for using WES to identify novel disease-causing mutations in individual patients.

Thank you for reviewing our manuscript and the very positive and useful comments.

Minor Concerns:

1. The authors should qualify their conclusions somewhat. For example, in the Discussion, they state: page 16, "to prove that it modifies the pathogenic impact on top of the bona fide SCN5A mutation"; and page 15, "TASK-4 as a new disease gene that is functionally relevant for cardiac conduction disorders." Although the evidence and arguments strongly support this likelihood, it remains formally possible that one of the other gene variants is actually responsible. That is, even though the algorithm and PPT analysis predicts no effect of those other variants, there is no direct experimental evidence to rule them out definitively. Also, there is no obvious way to "prove" that the TASK-4 mutation is actually responsible in this individual - the evidence is, by nature, circumstantial.

We agree that there is no direct experimental evidence to definitively rule out that other factors contribute to the phenotype and thus, de-emphasized some of our statements according to your suggestion. The statement on page 17 (former page 16) was changed to "to prove that it may modulate the pathogenic impact on top of the bona fide SCN5A mutation". Accordingly we de-emphasized our statements on page 16 (former page 15): "TASK-4 as a novel and potentially relevant disease gene that might be functionally relevant for cardiac conduction disorders."

In addition, we added a Discussion section on page 17, at the end of the first paragraph: "Although our experimental evidence strongly supports the likelihood that TASK-4 is a new disease gene that is functionally relevant for cardiac conduction disorders, it remains formally possible that one of the other gene variants is also relevant or contributing to the phenotype. Even though the algorithm and PPT analysis predicts no effect of those other variants, there is no direct experimental evidence to definitively rule out the role of other genes. As mice do not have a KCNK17 gene and thus cannot be used to develop a disease model for TASK-4 mutations, there is no obvious way to "prove" that the TASK-4 mutation is actually responsible in this individual - the evidence is, by nature, circumstantial. Nevertheless, the combination of our genetic data, the newly identified preferential expression of KCNK17 in the conduction system, together with the strong electrophysiological phenotype that we have identified, clearly suggests that the G88R mutation is a disease modifying mutation for PCCD."

2. It is not entirely certain that the currents generated from co-expression experiments reflect heterodimeric channels containing both a wild type and mutated subunit, rather than additive effects of distinct populations of homomeric channels comprising only wild type or mutated subunits. The quantitative analysis that predicted the same current amplitude for 20.83 ng injection may be simply fortuitous. In any case, it is not necessarily crucial for the overall interpretation whether or not the individual channels are truly heteromeric, so perhaps this conclusion should be tempered as well (e.g., see p. 13, and elsewhere).

Thank you for discussing this point. To further highlight that the gain-of-function is not just additive, we now include the data of the 12.5 ng G88R injection and provide a new Figure (Supplementary Figure 2 and Results page 13 (last line) and page 14 (first line)). In the novel Supplementary Figure 2, we highlight that the study was performed in a linear range, as injection of twice the amount of TASK-4 cRNA into *Xenopus* oocytes leads to a doubling of the current amplitude. Injection of 12.5 ng G88R leads to a 2.58 ± 0.2 fold increase in current amplitude compared to injection of 12.5 ng wild-type cRNA. This is a similar gain-of-function, as we have observed, when using 25 ng of cRNA for the constructs (Fig. 6B). Co-expression of 12.5 ng wild-type TASK-4 with 12.5 ng of G88R leads to a pronounced current increase, which is bigger than adding the amplitudes for both the individual constructs (Supplementary Figure 2; calculation no heteromers). Note that such an additive behavior would only occur if the channels would not form heteromers and express as separate homomeric channels. However, there is no evidence that the G88R should fail to form heteromeric channels with wild-type TASK-4, especially as our fluorescence imaging does not show a separate G88R population (Fig. 4D). Most importantly, after co-expression with wild-type and assuming a normal assembly, only 16.67 % of the channels would have two G88R subunits (Fig. 6C). Thus, if the gain-of-function would not be conferred to heteromeric channels with wild-type subunits, only 16.67 % of the dimeric channels would show a gain-of-function. The resulting current would be formed by 16.67 % of the G88R amplitude plus 83.33 % of wild-type amplitude. Calculating the expected current when only the channels with two G88R subunits have a gain-of-function (Supplementary Figure 2; calculation non-dominant) shows that the observed strong current increase by co-expression can only be explained if heteromeric channels of wild-type and G88R subunits also have a gain-of-function. Thus, it is for us the most straightforward interpretation that G88R is assembled with wild-type subunits which is conferring a gain-of-function to the heteromeric channel complex.

Nevertheless, as suggested we tempered our statements i.e. on page 14 “*Our data clearly showed that the G88R mutant acts in a dominant-manner*” was changed to “*The most straightforward interpretation of our data is, assuming a regular assembly that the G88R mutant acts in a dominant-manner*”; And the last sentence of the first paragraph on page 14 was changed to “*The dominant-active gain-of-function by the G88R exchange suggests that in heterozygous patients the majority of native cardiac TASK-4.....*”.

3. *It was surprising that no data were presented for TASK-4(G88R) at 12.5 ng in Fig. 6B. Please include those data.*

These data were initially not included, as the experiments were designed to mimic the most common clinical states, meaning wild-type (two healthy alleles), a haploinsufficiency (only one healthy allele), a heterozygous (one wild-type and one mutant allele) and a homocytotic state (two mutant alleles). Thus, we did initially not provide the current amplitudes for 12.5 ng of the G88R mutant, as it would reflect the more unlikely situation of a mutant allele in the presence of a haploinsufficiency. As we were working in a linear range, providing the current amplitudes of the 12.5 ng G88R cRNA injection did, to our initial opinion, not provide any additional information, as we just observed the similar gain-of-function (2.6-fold), as when recording the mutant with 25 ng (2.9-fold in Fig. 4B and 2.6-fold in Fig. 6B). However, in the context of the question that was raised above (comment 3), we now include this data in the novel Supplementary Figure 2 to highlight that the gain-of-function is conferred to the heteromeric channels and that the increased amplitudes are not caused by an additive effect.

4. *For the uninitiated reader, it would be helpful to provide some markup of the panels in Fig.1 that could highlight the key measures on the ECG and how they changed over the 5 year observation period.*

Thank you for this suggestion. In the revised manuscript, we have now included markups for the relevant segments in the ECGs.

Referee #3 (Comments on Novelty/Model System):

The manuscript by Friedrich and colleagues report a combined candidate gene and whole-exome sequencing strategy to determine the genetic basis for a severe case of combined progressive cardiac conduction system disease (PCCD) and idiopathic ventricular fibrillation (IVF) in a single adult male. They discovered a novel splice site mutation in the cardiac sodium channel gene (SCN5A) and a gain-of-function nonsynonymous variant in a twin-pore potassium channel (TASK-4) with a plausible, but unproven, role in cardiac electrophysiology.

The study main focuses on the functional consequences of the TASK-4 variant, which provide convincing evidence of gain of function. However, the main conclusions of the paper implicate TASK-4 as a genetic modifier of the phenotype, a conclusion that is severely weakened by the lack of any genotype-phenotype correlation. An effort to determine the ECG phenotypes of first degree relatives and the associated genotypes at SCN5A and TASK-4 might reveal additional evidence supporting their main claim. Further experiments in a myocyte system to understand the cellular consequences of a gain-of-function in TASK-4 would also strengthen the paper.

Referee #3 (Remarks):

The manuscript by Friedrich and colleagues report a combined candidate gene and whole-exome sequencing strategy to determine the genetic basis for a severe case of combined progressive cardiac conduction system disease (PCCD) and idiopathic ventricular fibrillation (IVF) in a single adult male. They discovered a novel splice site mutation in the cardiac sodium channel gene (SCN5A) and a gain-of-function nonsynonymous variant in a twin-pore potassium channel (TASK-4) with a plausible, but unproven, role in cardiac electrophysiology. The study main focuses on the functional consequences of the TASK-4 variant, which provide convincing evidence of gain of function. However, the main conclusions of the paper implicate TASK-4 as a genetic modifier of the phenotype, a conclusion that is severely weakened by the lack of any genotype-phenotype correlation.

We thank Referee #3 for these clear comments and for reviewing our manuscript.

Major comments

1. Addition genotype-phenotype data are required to provide a convincing argument that TASK-4 modifies the trait. An effort to determine the ECG phenotypes of first degree relatives and the associated genotypes at SCN5A and TASK-4 is essential. It remains possible that TASK-4 is unrelated to the phenotype.

We absolutely agree that having a genotype-phenotype correlation for the family would substantially strengthen the manuscript and support our experimental conclusions. As stated for Referee #1, we did a lot of personal efforts to convince other family members, in particular first degree relatives to participate in our study. However, the fate of the index patient closed the door for others to participate - probably in the light of anxiety and potential recurrence of cardiac events in relatives. However, we were able to reconstruct the family history, at least, and already introduced this into the original manuscript (page 7). Here we previously noted *"The family history was negative for sudden cardiac death or known inherited cardiac conditions"*. In addition, we now explicitly state on page 10: *"Since DNA from other family members was not available, we were not able to proof whether the identified genetic mutations in both genes were inherited or occurred as de-novo ones, however the family history was not further indicative for other arrhythmias or sudden cardiac death"*.

Nevertheless, our study is as we think, an excellent example that the novel technique of whole exome sequencing can provide answers in single arrhythmia cases, where in former days classical genetics would have failed. In the revised manuscript we now carefully discuss the problems arising by identifying a novel arrhythmia gene in a single index patient utilizing whole exome sequencing (page 17, end of the first paragraph): *"Although our experimental evidence strongly supports the likelihood that TASK-4 is a new disease gene that is functionally relevant for cardiac conduction disorders, it remains formally possible that one of the other gene variants is also relevant or contributing to the phenotype. Even though the algorithm and PPT analysis predicts no effect of those other variants,*

there is no direct experimental evidence to definitively rule out the role of other genes. As mice do not have a KCNK17 gene and thus cannot be used to develop a disease model for TASK-4 mutations, there is no obvious way to "prove" that the TASK-4 mutation is actually responsible in this individual - the evidence is, by nature, circumstantial. Nevertheless, the combination of our genetic data, the newly identified preferential expression of KCNK17 in the conduction system, together with the strong electrophysiological phenotype that we have identified, clearly suggests that the G88R mutation is a disease modifying mutation for PCCD."

2. Additional evidence should be provided to support that a gain-of-function variant in TASK-4 will affect cardiomyocyte resting potential or some other cellular electrophysiological phenotype. Rather than hyperpolarize the resting membrane potential, this variant might hinder or slow the upstroke of an action potential by requiring a strong depolarization to overcome the leak channel's effect. This information would be extremely valuable for supporting the author's main conclusions.

Since TASK-4 is not expressed in mice, it is not possible to develop a transgenic G88R mouse as a disease model for PCCD. As TASK-4 is preferentially expressed in the conduction system, transfection of G88R into ventricular cardiomyocytes would not provide the necessary mechanistic information to explain the effects of the mutation on conductivity. HL-1 cells are spontaneously beating sino-atrial node like cardiomyocytes. As these are more closely related to cells in the conduction system, we performed additional experiments using this cell type. As Referee 3 already anticipated, the gain-of-function by G88R, as compared to the overexpression of TASK-4 in HL-1 cells, does not only cause a hyperpolarization, but also antagonizes depolarization. This can be noted by a strong slowing of the upstroke velocity. In addition, G88R induces a shortening of the action potential duration, a hyperpolarization following the action potential and a long phase of diastolic depolarization, resulting in a reduced action potential frequency of the spontaneously beating HL-1 cells.

The novel data has been introduced in the Results section on page 14-15 of the revised manuscript, the new Figure 7 and Figure Legend, the Methods section and the Supporting Information Movies S1 to S3.

We agree that the requested mechanistic experiments were extremely valuable for supporting our main conclusions. As in the original version of the manuscript, we propose that a stabilization of the membrane potential in the conduction system might lead to a slowed conductivity, but now we also highlight that a slowed upstroke velocity in the conduction system might contribute to the phenotype of PCCD. In addition, we hope that the re-discussion of our data in our point-by-point response address the remaining concerns raised by Referee 3.

In the revised Results section we have also included a quote of the cellular effects you proposed above. The novel Results section now reads:

"G88R mutants stabilize the membrane potential and slow upstroke velocity of spontaneously beating HL-1 cells

Since TASK-4 is not expressed in mice, it is not possible to develop a transgenic 'G88R mouse' as a disease model for PCCD. As we found that TASK-4 is preferentially expressed in the conduction system, transfection of G88R into ventricular cardiomyocytes would not provide sufficient mechanistic information to explain the effects of the mutation on conductivity. HL-1 cells are spontaneously beating sino-atrial node like cardiomyocytes (Claycomb et al, 1998) and as these are more closely related to cells in the conduction system, we performed additional experiments using this cell type. We transfected EGFP-tagged wild-type TASK-4 or G88R in HL-1 cells and measured action potential frequency of the spontaneously beating HL-1 cells (Fig 7A and Supporting Information Movies S1 to S3) and characterized the action potentials using patch clamp experiments (Fig 7C-J). Transfection of wild-type TASK-4 into HL-1 cells already slowed the action potential frequency from 179 ± 4 bpm to 125 ± 2 bpm (Fig 7A and Supporting Information Movies S1 and S2), as expected for the overexpression of a tandem K^+ channel in cells with a less hyperpolarized membrane potentials, as in the sino-atrial node

or in the conduction system. Most importantly, transfecting the same amount of G88R TASK-4 cDNA, with a similar efficiency and similar protein expression (Fig 7B), caused a significantly more pronounced slowing of the spontaneous beating frequency (Fig 7A and Supporting Information Movies S2) and the frequency was reduced to 59 ± 3 bpm. In patch clamp recordings the action potential frequency of untransfected HL-1 cells was much slower (Fig 7D), presumably reflecting the lack of supplemented Claycomb media which for instance contains norepinephrine. However, even under these non-stimulated conditions, the action potential frequency, recorded in the current clamp mode, of G88R transfected cells was much slower than that of TASK-4 transfected cells (Fig 7C and D). In addition, the patch clamp experiments showed that the overexpression of G88R, compared to TASK-4, leads to a significantly more pronounced shortening of the action potential duration (Fig 7C and E), while the maximal diastolic membrane potential is more hyperpolarized (Fig 7C). This effect by G88R can be quantified by a more pronounced afterhyperpolarisation following the action potential (Fig 7F and G). Overexpression of TASK-4 and G88R also antagonizes depolarization, which can be noted by a reduced action potential overshoot (Fig 7H and I) and a strong slowing of the upstroke velocity (Fig 7H and J). While the reduction of the action potential overshoot was already fully achieved by the overexpression of wild-type TASK-4 (Fig 7H and I), the gain-of-function by G88R caused a much more pronounced slowing of the upstroke velocity (Fig 7H and J). In summary, these overexpression experiments demonstrate that G88R leads to similar, but much stronger effects than the overexpression of wild-type TASK-4. Our data indicate that wild-type TASK-4 can hyperpolarize the resting membrane potential of cells in the conduction system and that the G88R mutation might hinder or slow the upstroke of an action potential by requiring a strong depolarization to overcome the leak channel's effect.

Thus, we propose that a stabilization of the membrane potential in the conduction system by G88R and especially a slowed upstroke velocity in the conduction system might contribute to the phenotype of slowed conductivity in PCCD.”

Minor concerns

a. Some references appear as numbers in the text (see page 15, end of first paragraph for example).

Thank you. We included the reference (page 16 of the revised manuscript).

b. Page 17, 3rd line from bottom - the parenthetical '= pseudogene' has an obscure meaning. Please explain what you mean.

The term `pseudogene` on former page 17 was removed and we now state on page 19: “As there are no specific TASK-4 blockers available and mice do not functionally express a KCNK17 gene”.

2nd Editorial Decision

21 May 2014

Thank you for the submission of your revised manuscript to EMBO Molecular Medicine. We have now received the enclosed reports from the referees that were asked to re-assess it. As you will see the reviewers are now globally supportive and I am pleased to inform you that we will be able to accept your manuscript pending the following final amendments:

1) please correct your manuscript as suggested by referee 1

Please submit your revised manuscript within two weeks. I look forward to seeing a revised form of your manuscript.

***** Reviewer's comments *****

Referee #1 (Comments on Novelty/Model System):

This paper addresses a clinically relevant topic with a good animal model system. The data are clear and demonstrated a novel finding - specifically that IgE can contribute to T cell and macrophage activation, contributing to AAA.

Referee #1 (Remarks):

The authors submitted an exhaustive rebuttal, and made considerable efforts to address my comments. I appreciate their sincerity in this. I have only a few minor issues:

1. The term FceRI is used, rather than FcεR1. I have always seen the use of the roman numeral. It is fine with me if the authors don't want to edit every figure, but changing this in the text is recommended.
2. Figure 1F is labeled as 1E in the legend.
3. The term LgE is used, instead of IgE, in the legend to Figure 2B.

Referee #2 (Remarks):

All issues have been addressed.

Referee #3 (Remarks):

This reviewer thanks the authors for diligently addressing the concerns raised during my evaluation of the original version of the manuscript.

All of my concerns / questions have been satisfactorily addressed / answered with the revised version.

2nd Revision - authors' response

21 May 2014

Point-by-Point Responses to Reviewer #1

We thank this reviewer for his/her thorough evaluations of our prior submissions. We deeply appreciate his/her time on our manuscript. *Here are few his/her remaining suggestions/corrections*

that we highlighted in bold and italic, followed by our responses.

1. The term FceRI is used, rather than FceR1. I have always seen the use of the roman numeral. It is fine with me if the authors don't want to edit every figure, but changing this is the text is recommended.

We have converted FceR1 into FceR1a in revised Figure 8, and made sure all other Figures used FceR1a.

2. Figure 1F is labelled as 1E in the legend.

We have made the correction.

3. The term LgE is used, instead of IgE, in the legend to Figure 2B.

We have made the correction.